# Cardiolipin targets a dynamin-related protein to the nuclear membrane

**Usha Pallabi Kar[†], Himani Dey[†], Abdur Rahaman***

School of Biological Sciences, National Institute of Science Education and Research-HBNI, Bhubaneswar, India

**Abstract** Dynamins are targeted to specific cellular membranes that they remodel via membrane fusion or fission. The molecular basis of conferring specificity to dynamins for their target membrane selection is not known. Here, we report a mechanism of nuclear membrane recruitment of Drp6, a dynamin member in *Tetrahymena thermophila*. Recruitment of Drp6 depends on a domain that binds to cardiolipin (CL)-rich bilayers. Consistent with this, nuclear localization of Drp6 was inhibited either by depleting cellular CL or by substituting a single amino acid residue that abolished Drp6 interactions with CL. Inhibition of CL synthesis, or perturbation in Drp6 recruitment to nuclear membrane, caused defects in the formation of new macronuclei post-conjugation. Taken together, our results elucidate a molecular basis of target membrane selection by a nuclear dynamin and establish the importance of a defined membrane-binding domain and its target lipid in facilitating nuclear expansion.

**\*For correspondence:**
arahaman@niser.ac.in

[†]These authors contributed equally to this work

**Competing interests:** The authors declare that no competing interests exist.

## Introduction

Topological changes and remodeling of membranes are fundamental processes in cellular physiology. Intricate biological machineries have evolved to facilitate these changes in living cells. Dynamins and dynamin-related proteins (DRPs) comprise a family of large GTPases that catalyze membrane remodeling reactions (*Praefcke and McMahon, 2004*). Members of the dynamin family are mechano-chemical enzymes that couple the free energy of GTP hydrolysis with membrane deformation, thereby performing important cellular functions ranging from scission of membrane vesicles, cytokinesis, and maintaining mitochondrial dynamics to conferring innate antiviral immunity (*Ramachandran and Schmid, 2018*). The common features shared by all dynamins and DRPs are the presence of a large GTPase domain (GD), a middle domain, and a GTPase effector domain (GED), which distinguish them from other GTPases (*Praefcke and McMahon, 2004*). The MD and the GED are involved in oligomerization and regulation of GTPase activity (*Ramachandran et al., 2007*). The feature that distinguishes DRPs from classical dynamins is the lack of a defined pleckstrin homology domain (PH domain) and a proline-rich domain.

All dynamin proteins undergo rounds of assembly and dis-assembly on the target membrane and tubulate the underlying membrane, which is required for fission or fusion function. The members of this family become associated with lipids on the target membrane and are important for performing cellular functions (*Ramachandran and Schmid, 2018*). The PH domain in classical dynamins binds to phosphatidyl inositol 4,5 bis-phosphate (PIP2) at the target sites of endocytosis and plays an essential role in vesicle scission during endocytosis (*Zheng et al., 1996*). All the DRPs lack PH domains and instead possess either lipid binding loops or trans-membrane domains for membrane recruitment or association (*Ramachandran and Schmid, 2018*). A stretch of positively charged amino acid residues in the lipid binding loops of all known DRPs interacts with the negatively charged head groups of the lipids, and this interaction is important for target membrane association (*Rujiviphat et al., 2009*; *von der Malsburg et al., 2011*; *Bustillo-Zabalbeitia et al., 2014*; *Smaczynska-de Rooij et al., 2019*; *Wang et al., 2019*; *Yan et al., 2020*).

Nuclear remodeling including its expansion is a fundamental process in eukaryotes, the mechanism of which is not well understood in any organism. It requires remodeling of nuclear membrane and incorporation of new materials including lipids and proteins into the existing membrane. *Tetrahymena thermophila*, a unicellular ciliate, exhibits nuclear dimorphism. Each cell harbors a small, diploid, transcriptionally inactive micronucleus (MIC) and a large, polyploid, transcriptionally active macronucleus (MAC) (*Karrer, 2000*). During sexual conjugation in *Tetrahymena*, two cells of complementary mating types first form a pair, followed by a series of complex nuclear events resulting in the loss of the parental MAC. Subsequently, new MIC and MAC are formed through the fusion of haploid nuclei produced from parental micronuclei. The precursors of the new MIC and MAC are identical in size and in genome content at the initial stage of nuclear differentiation. However, two of the four new MICs rapidly enlarge, making the final volume 10- to 15-fold larger, and develop into MACs (*Cole et al., 1997*). This process calls for a sudden and dramatic expansion of nuclear envelope. Drp6, which is one of the eight DRP paralogs in *Tetrahymena*, is specifically upregulated when the MICs rapidly expand to form new MACs. Drp6 associates with outer nuclear envelope of both MIC and MAC (*Elde et al., 2005*), and inhibition of Drp6 function results in a profound deficiency in the formation of new MACs (*Rahaman et al., 2008*). It has been recently demonstrated that Drp6 functions as an active GTPase and self-assembles into higher order helical spirals and ring structures, and therefore resembles other members of the family (*Kar et al., 2018*).

In the present study, we have elucidated a mechanism for Drp6 recruitment in the nuclear membrane. Our results reveal that Drp6 directly interacts with membrane lipids and that CL acts as its physiologically important target lipid. Furthermore, we have identified a lipid-binding domain in Drp6 and provide evidence that the domain plays a pivotal role in nuclear association of Drp6 and therefore in nuclear expansion.

## Results

### A DRP-targeting determinant is important for nuclear recruitment of Drp6

All the known dynamin family proteins perform cellular functions by associating with a target membrane. Classical dynamins contain a pleckstrin homology (PH) domain responsible for membrane binding (*Figure 1a*). Drp6 associates with nuclear envelope and regulates nuclear remodeling in *Tetrahymena*. However, Drp6 like other DRPs lacks a PH domain or any recognizable membrane-binding domain (*Figure 1a*). Sequence alignment revealed the presence of a region in Drp6 that is located at the position of the PH domain of classical dynamin (*Figure 1b*, *Figure 1—figure supplement 1a*) and that was earlier named the Drp-targeting determinant (DTD) (*Elde et al., 2005*).

To assess whether the DTD is important for recruitment of Drp6 to target membranes, full-length *DRP6* and *drp6ΔDTD* were expressed separately as N-terminal GFP-fusion proteins and their cellular distributions were compared by confocal microscopy. As expected, Drp6 was chiefly present on the nuclear envelope and also on some cytoplasmic puncta (*Figure 1c*, *Figure 1—figure supplement 1b*). These cytoplasmic puncta of Drp6 are endoplasmic reticulum derived vesicles (*Rahaman et al., 2008*). When confocal images of *GFP-drp6ΔDTD*-expressing cells were analyzed, it was observed that the deletion of DTD resulted in complete loss of nuclear localization, and it was mainly associated with cytoplasmic puncta (*Figure 1c*, *Figure 1—figure supplement 1b*). These results clearly demonstrate that DTD is necessary for recruiting Drp6 to the nuclear envelope, but not to cytoplasmic puncta. We next evaluated if the DTD is sufficient for nuclear envelope targeting, by expressing it as a GFP-fusion protein. The confocal images of cells expressing GFP-*drp6-DTD* showed that GFP-DTD often appeared as cytoplasmic puncta, but that nuclear envelope localization in a subset of cells is detectable albeit less prominently as compared to that of GFP-drp6 (*Figure 1c*, *Figure 1—figure supplement 1b*). This suggests that the DTD is able to interact with the nuclear envelope. This interaction appears weaker than for the full-length protein, suggesting that other domains also contribute to nuclear recruitment of Drp6. Similarly, other dynamin family members rely for their targeting on a membrane-binding domain but also depends on other domains such as GD (*Vallis et al., 1999*; *von der Malsburg et al., 2011*; *Bramkamp, 2012*). We conclude that DTD is essential but not sufficient for Drp6 recruitment to the nuclear envelope. In contrast, Drp6 does not require its DTD for targeting to the ER-derived cytoplasmic vesicles.

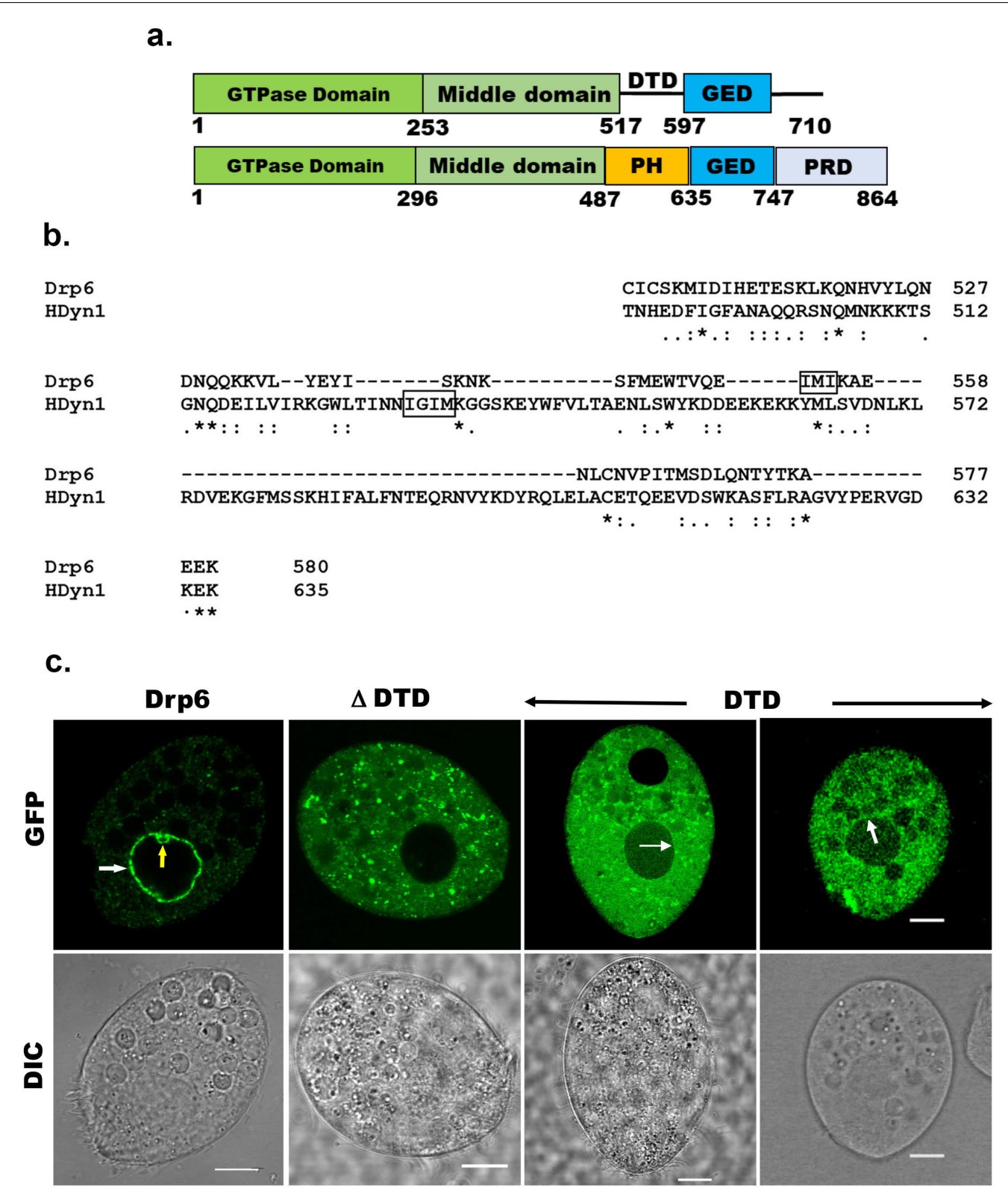

**Figure 1.** Identification of the region of Drp6 important for nuclear recruitment. (a) Diagram showing domains of *Tetrahymena* Drp6 (top) and human dynamin 1 (bottom). Five domains of dynamin indicated as GTPase domain, middle domain, PH domain, GED, and PRD. Drp6 contains three domains but lacks PH domain and PRD. Numbers indicate the position of amino acids in the protein. (b) Sequence alignment of *Tetrahymena* dynamin-related protein 6 (Drp6) and human dynamin 1 (HDyn1) generated using Clustal Omega. Only the PH domain of HDyn1 and the corresponding aligned region

*Figure 1 continued on next page*

*Figure 1 continued*

of Drp6 are shown. The hydrophobic patch (IGIM) of PH domain important for membrane insertion is shown within a box. A putative hydrophobic patch (IMI) in Drp6 is also within box. (**c**) Confocal images of live *Tetrahymena* cells expressing GFP-drp6 (Drp6), GFP-drp6ΔDTD (ΔDTD), and GFP-drp6-DTD (DTD) are shown. While localization on the nuclear envelope of MAC is indicated by white arrow, yellow arrow indicates MIC. Bar = 10 μm.

The online version of this article includes the following figure supplement(s) for figure 1:

**Figure supplement 1.** DTD is important for nuclear localization of Drp6.

## DTD is the membrane-binding domain of Drp6

Recruitment of a protein to a target membrane is achieved either by interaction with membrane lipids or by forming a complex with another membrane protein. The DTD is essential for recruitment of Drp6 to the nuclear envelope and may represent a membrane-binding domain. To test this idea, we generated N-terminal histidine-tagged *drp6-DTD* and *DRP6* for bacterial expression and purification and then used the purified proteins in lipid overlay assays. Drp6 was purified to near homogeneity (*Figure 2a*) and incubated with total *Tetrahymena* lipid spotted on nitrocellulose membrane either in the presence or absence of GTP. Drp6 interacts with *Tetrahymena* lipid with or without GTP (*Figure 2b*), suggesting that it harbors a lipid-binding domain. To identify the lipids with which Drp6 interacts, we performed the overlay assay using commercially available strips spotted with fifteen different lipids (Echelon Biosciences, USA). The results demonstrated that Drp6 specifically interacts with three phospholipids, namely phosphatidylserine (PS), phosphatidic acid (PA), and cardiolipin (CL) (*Figure 2b*). In order to find out whether lipid binding is a property of the DTD, we partially purified DTD as an N-terminal his tagged protein (*Figure 2a*) and used it for the overlay assay. Like the full-length protein, DTD also interacts with all three phospholipids (*Figure 2b*).

We then looked at lipid binding in the physiologically relevant context of a bilayer, using an *in vitro* binding assay to liposomes containing 10% PA, 10% PS, or 10% CL. All the liposomes also contained 70% PC and 20% PE. In sucrose density gradients, the recombinant Drp6 co-migrated with all the three types of liposomes, appearing in the top (light) fractions (*Figure 2c*). Drp6 was not found in the gradient top fractions in the absence of added liposomes (*Figure 2c*). Similarly, Drp6 also failed to co-migrate with liposomes that contained only PC and PE (*Figure 2c*). Taken together, our results indicate that Drp6 interacts with membranes *in vitro* and that this depends on the presence of either PS, PA, or CL. DTD by itself interacts with the same three phospholipids as holo-Drp6, consistent with the idea that DTD is the membrane-binding domain. To further test this idea, we performed sub-cellular fractionation of cells expressing *GFP-drp6-DTD* or *GFP-drp6*. As shown in *Figure 2d*, Drp6 appeared in both soluble and membrane fractions. DTD also appeared in the membrane fraction, suggesting that it can bind to membranes *in vivo* (*Figure 2d*). Taken together, these results lead us to conclude that DTD is a membrane-binding domain and requires PS, PA, or CL for association with the membrane.

## A single-point mutation (I553M) in the membrane-binding domain abrogates nuclear recruitment of Drp6

Dynamin binds to membrane lipids via interaction between its PH domain and the PIP2 head group. A hydrophobic patch in the PH domain of classical dynamin is important for membrane association/insertion (*Ramachandran et al., 2009*). The sequence similarity between PH domain of human dynamin 1 and corresponding region of Drp6 is very low (*Figure 1b*). However, the structural similarity is very high among all the dynamin family proteins whose structures are known. Therefore, we generated a three-dimensional model of Drp6 in order to identify the corresponding hydrophobic region in the DTD. The 3-D modeling of Drp6 shows that the structure of DTD is not related to PH domain, but that nonetheless has a hydrophobic patch (aa 553–555) (*Figure 3a*). To test the importance of this patch, we substituted the first residue, I553, with M. We expressed this mutant allele as an N-terminal GFP-fusion (GFP-Drp6-I553M), and in parallel expressed a GFP-fusion of the wild-type protein (GFP-drp6), and characterized their localization in *Tetrahymena* by confocal microscopy. While the latter localized mainly on the nuclear envelope with few cytoplasmic puncta, the mutant GFP-drp6-I553M was not visible on the nuclear envelope but was instead exclusively present at cytoplasmic puncta (*Figure 3b*, *Figure 3—figure supplement 1*). This difference suggests that the isoleucine at 553rd position is important for nuclear localization of Drp6.

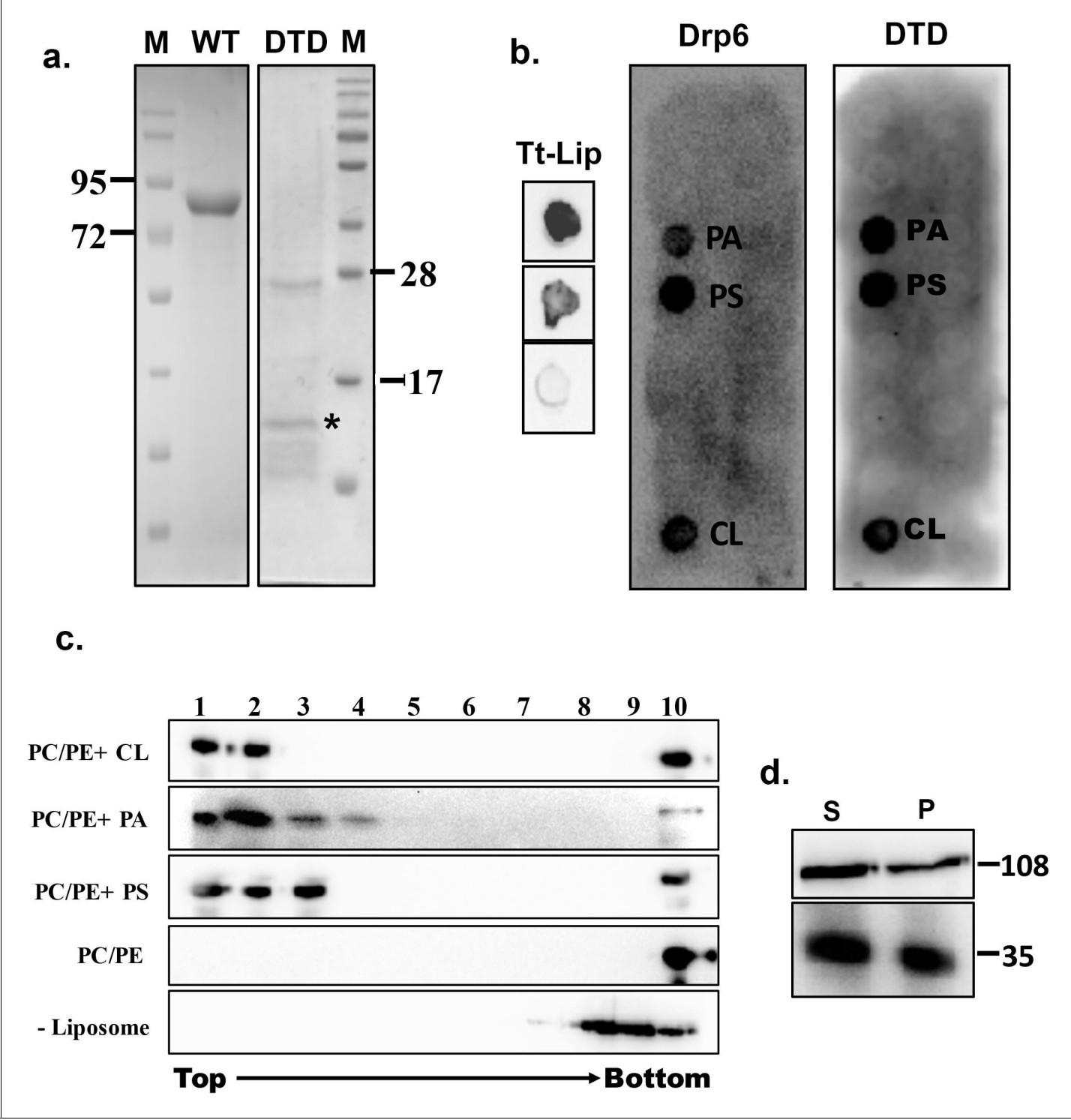

**Figure 2.** Identification of membrane-binding domain of Drp6. (a) Coomassie-stained SDS–PAGE gels showing purification of His-drp6 (WT) and His-drp6-DTD (DTD) expressed in *Escherichia coli*. M is the molecular weight markers. Some of the markers are indicated on the sides. The purification of His-drp6-DTD was partial and contained additional proteins from *E. coli*, including one prominent band below 28 kDa. The purified His-drp6-DTD appearing below 17 kDa marker is indicated by an asterisk. (b) Lipid overlay assay as detected by western blot analysis using anti-his antibody. (Tt-Lip); total *Tetrahymena* lipid spotted on nitrocellulose membrane and incubated with His-drp6 in absence (top) or presence (middle) of GTP. The bottom spot is incubated with BSA. Strip spotted with 15 different lipids and incubated either with His-drp6 (Drp6) or with His-drp6-DTD (DTD). Both Drp6 and DTD interacted with PA, PS, and CL are indicated. (c) Floatation assay using liposomes containing 70% phosphatidylcholine and 20% phosphatidylethanolamine additionally supplemented with 10% CL (PC/PE+CL), 10% PA (PC/PE+PA), or 10% PS (PC/PE+PS). While liposomes in (PC/

*Figure 2 continued*

PE) contained 80% phosphatidylcholine and 20% phosphatidylethanolamine, no liposome was added in (–Liposome). His-drp6 was incubated either with different liposomes or without liposomes, overlaid with sucrose gradient, and subjected to ultra-centrifugation. Fractions were collected from top and detected by western blot analysis using anti-his antibody. Drp6 appearing in the top four fractions indicate interaction with liposome. The experiments were repeated at least three times, and representative results are shown here. (d) Lysates of *Tetrahymena* cells expressing either GFP-drp6 (top) or GFP-drp6-DTD (bottom) were fractionated into soluble (S) and membrane (P) fractions and detected by western blot using anti-GFP antibody. Molecular weights of the proteins are indicated on the right.

## Mutation at I553 does not affect GTPase activity and self-assembly of Drp6

Dynamin and DRPs require binding and hydrolysis of GTP for target membrane localization and membrane remodeling functions. Drp6 hydrolyses GTP *in vitro* (*Kar et al., 2018*). To understand whether the mutation at I553 affects GTP binding and/or hydrolysis, we expressed and purified Drp6-I553M and Drp6 as N-terminal histidine-tagged proteins (*Figure 4a*) and compared their GTPase activities. The GTPase activity of Drp6-I553M (0.061 ± 0.002 nmol/µM/min) was not significantly different from that of Drp6 (0.056 ± 0.003 nmol/µM/min) (*Figure 4b*). The GTPase activities of both wild-type and mutant proteins were also found to be similar when reactions were carried out for 0–20 min (*Figure 4b*). We also assessed the Michaelis–Menten constant ($K_m$) and maximum velocity ($V_{max}$) for both these proteins (*Figure 4c*). The $K_m$ of Drp6-I553M (384 µM) was found to be slightly higher than that of Drp6 (180 µM), suggesting a marginal decrease in GTP binding affinity. The mutation did not affect $V_{max}$ (0.089 nmol/µM/min for Drp6; 0.0893 nmol/µM/min for Drp6-I553M). Taken together, these results suggest that the defect in nuclear localization of Drp6-I553M is not due to defective GTPase activity.

Another property of DRPs is their ability to assemble and dis-assemble at their target membranes. This involves self-assembly of helical spirals and ring structures and is important for membrane association and membrane remodeling functions. The self-assembly of Drp6 and Drp6-I553M was evaluated by size exclusion chromatography. Drp6 eluted in the void volume as an oligomer containing at least six monomers, as previously observed (*Kar et al., 2018*; *Figure 4d*). Similarly, Drp6-I553M also formed higher order structures, as the majority of the protein eluted in the void volume (*Figure 4d*). A small peak of material eluting near the 150 kDa marker might correspond to a dimer. The breadth of this smaller peak suggests that it consists of a mixture of monomeric and oligomeric structures, with a dimer at the peak fraction. Since the mutant protein was able to form higher order oligomeric structure, we suggest that mutation does not inhibit its self-assembly. However, there might be a difference in the assembly products formed by the wild-type and mutant proteins. To examine this possibility, we compared the ultrastructure of the wild-type and mutant proteins by electron microscopy of negatively stained preparations of the purified recombinant proteins. The Drp6 appeared mostly as large helical spirals, which are similar to structures found in other DRPs (*Figure 4e*). Ring-like structures were also present, as also found in other members of the family. Similarly, in Drp6-I553M samples, we observed both helical spirals and ring-like structures (*Figure 4e*). These results suggest that the I553M mutation does not block *in vitro* oligomerization of Drp6. Taken together, our results suggest that defective nuclear localization of Drp6-I553M is not due to defects in GTPase activity or perturbation of self-assembly.

## CL is important for nuclear recruitment of Drp6

Dynamin family proteins including Drp6 associate with their target membranes by interacting with specific lipids. To ask whether the I553M mutation might affect these interactions, we used *in vitro* membrane-binding assays. Recombinant Drp6 and Drp6-I553M proteins were incubated separately with liposomes containing either 10% CL, or PS, or PA. The association of the proteins with liposomes was then judged based on their co-flotation in sucrose density gradients. As can be seen in *Figure 5a*, Drp6 co-floated with all the three liposomes and appeared on the top fractions, whereas Drp6-I553M co-floated with PS- or PA-containing liposomes but failed to co-float with CL-containing liposomes. Since the mutant retains the ability to associate with PS- and PA-containing liposomes, the overall membrane-binding activity of Drp6 does not depend on I553. Instead, the mutation of I553 specifically inhibits interaction with CL. Based on these results, we infer that I553 in the

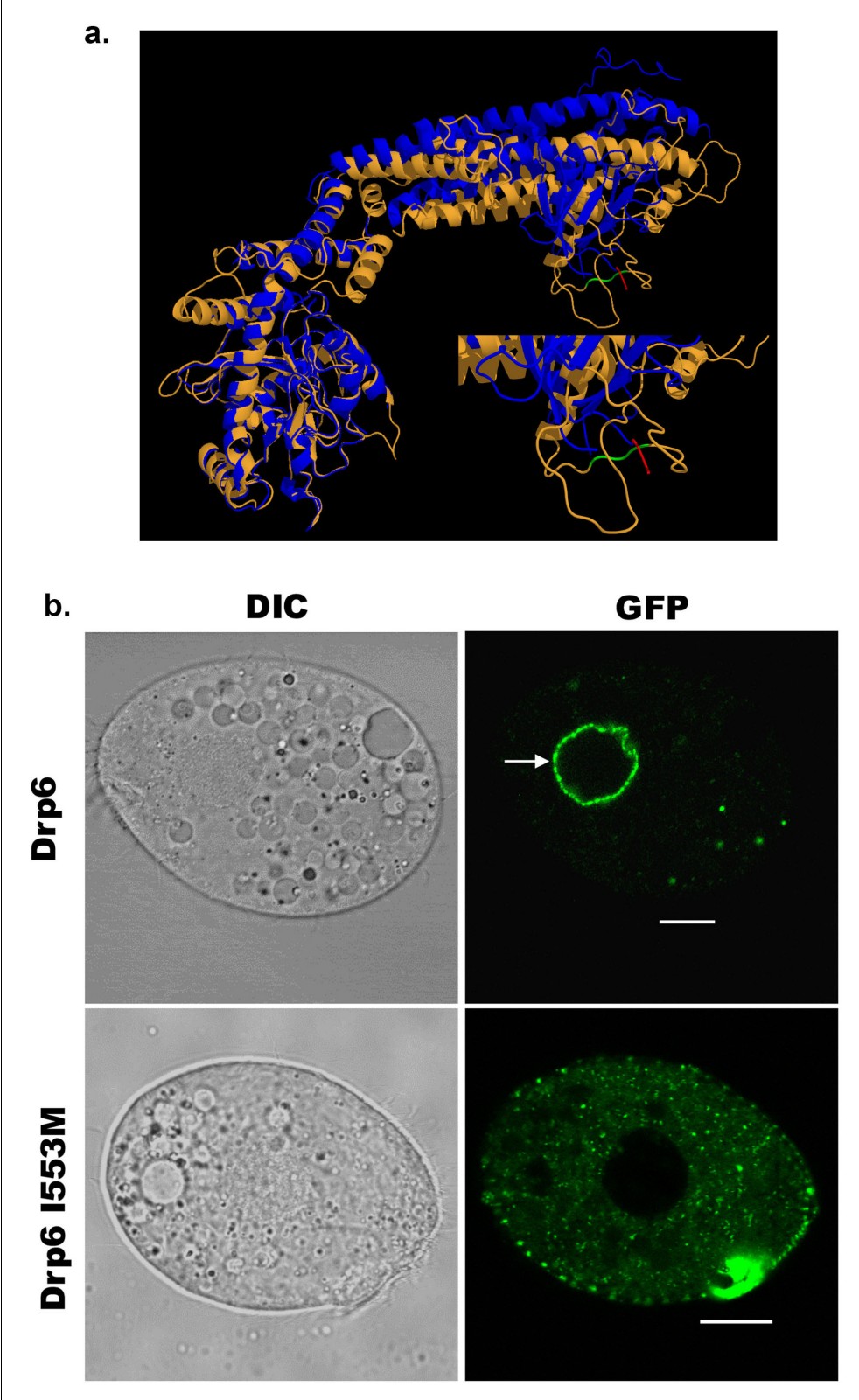

**Figure 3.** An isoleucine in the membrane-binding domain is important for nuclear localization of Drp6. (**a**) Three-dimensional structure of Drp6. Homology model of Drp6 (brown) was generated by I-TASSER using Human Dynamin-1 as template (blue). The part containing the hydrophobic patch (residues 531–534 marked in red) in the PH domain of Human Dynamin-1 important for membrane insertion along with the putative hydrophobic patch (residues 553–555 marked in green) of Drp6 model are shown at the bottom right after enlarging the area. Although far apart in primary sequences, the

*Figure 3 continued on next page*

*Figure 3 continued*

regions containing hydrophobic patch in both the proteins come to the vicinity in 3-D structure. (b) Confocal images of live *Tetrahymena* cells expressing GFP-drp6 (top) and GFP-drp6-I553M (bottom). Mutation of isoleucine to methionine at 553rd position leads to loss of nuclear localization. Arrow indicates nuclear envelope. Bar = 10 μm.

The online version of this article includes the following figure supplement(s) for figure 3:

**Figure supplement 1.** Confocal images of fixed *Tetrahymena* cells after DAPI-staining either expressing GFP-drp6 (Drp6) or GFP-drp6-I553M (Drp6-I553M).

membrane-binding domain is necessary for CL interactions and that these interactions are important for membrane targeting *in vitro*.

We then asked whether the interaction between Drp6 and CL was important for nuclear targeting *in vivo*. Importantly, while the nuclear envelope of animals (*Keenna et al., 1970*; *Kleinig et al., 1971*; *Sato et al., 1972*; *Jarasch et al., 1973*) lacks CL, it is present in the nuclear membrane of *Tetrahymena* (*Nozawa et al., 1973*). To evaluate whether CL is required for nuclear localization of Drp6, we depleted CL from *GFP-drp6*-expressing cells using pentachlorophenol (PCP), a polychlorinated aromatic compound. PCP is a respiratory uncoupler and a potent inhibitor of CL synthesis (*Ono and White, 1971*). Within 30 min of PCP treatment, GFP-drp6 dissociated from nuclear envelopes in the majority of cells while remaining associated with cytoplasmic puncta (*Figure 5b*). Quantitative analysis showed that while GFP-drp6 was localized at the nuclear envelope of all untreated cells, more than 80% of the cells completely lost nuclear localization of GFP-drp6 with the remaining cells showing decreased nuclear localization upon PCP treatment. To check that the nuclear envelope itself remains intact under these conditions, we localized the nuclear pore protein GFP-Nup3/MacNup98B. We found that GFP-Nup3/MacNup98B was clearly associated with nuclear envelopes before or after PCP treatment (*Figure 5b*). Moreover, PCP treatment does not disrupt membrane structure in general since the distribution of a cortical membrane-binding protein GFP-Nem1D (*Shukla et al., 2018*) was also not affected by this treatment (*Figure 5b*). These results therefore suggest that the delocalization of GFP-drp6 upon PCP treatment is due to loss of CL.

If the defect in localization of Drp6-I553M is due to a defect in CL-dependent targeting, one might expect that the defect would be suppressed in the presence of the wild-type protein, since the mutant protein would co-assemble with the correctly targeted wild type. To test this idea, we co-expressed *GFP-drp6-I553M* and *mCherry-drp6*. mCherry-drp6 colocalized almost entirely with GFP-drp6-I553M and was targeted to nuclear envelopes as well as cytoplasmic puncta, strongly suggesting that the mutant protein is able to co-assemble with the wild-type protein (*Figure 5c*). This result also reinforces the earlier conclusion that the mutation does not affect the overall structure of the protein.

## Interaction of I553 residue with CL determines nuclear recruitment of Drp6

After establishing the necessity of the I553 residue and CL in Drp6 nuclear recruitment, we addressed if the loss of nuclear recruitment of Drp6-I553M is specifically due to the presence of methionine at this position. To test this, we replaced I553 with alanine in the GFP-drp6-I553A protein and expressed it in the cell. Confocal microscopic analysis demonstrated that, similar to GFP-drp6-I553M, GFP-drp6-I553A also lost its association with nuclear envelope and was detected mostly as cytoplasmic puncta (*Figure 6a*, *Figure 6—figure supplement 1*). As expected, the wild-type GFP-drp6 was chiefly localized in the nuclear envelope under the same experimental conditions. This result reiterates our earlier conclusion that the I553 residue is critical for nuclear recruitment of Drp6. We further confirmed that the loss of Drp6-I553A recruitment to the nuclear envelope is specifically due to loss of its interaction with CL by performing floatation assay (*Figure 6b*). For this purpose, we purified recombinant Drp6-I553A protein and tested its binding to liposomes containing either CL, PS, or PA. Western blot analysis showed that Drp6-I553A lost interaction with CL liposomes while retaining its interaction with PS and PA liposomes (*Figure 6b*). To rule out that the loss of nuclear association is not due to the defect in GTPase activity or self-assembly, we performed size exclusion chromatography and GTP hydrolysis assay. When the purified recombinant Drp6-I553A protein was subjected to size exclusion chromatography, it eluted primarily in the void volume as

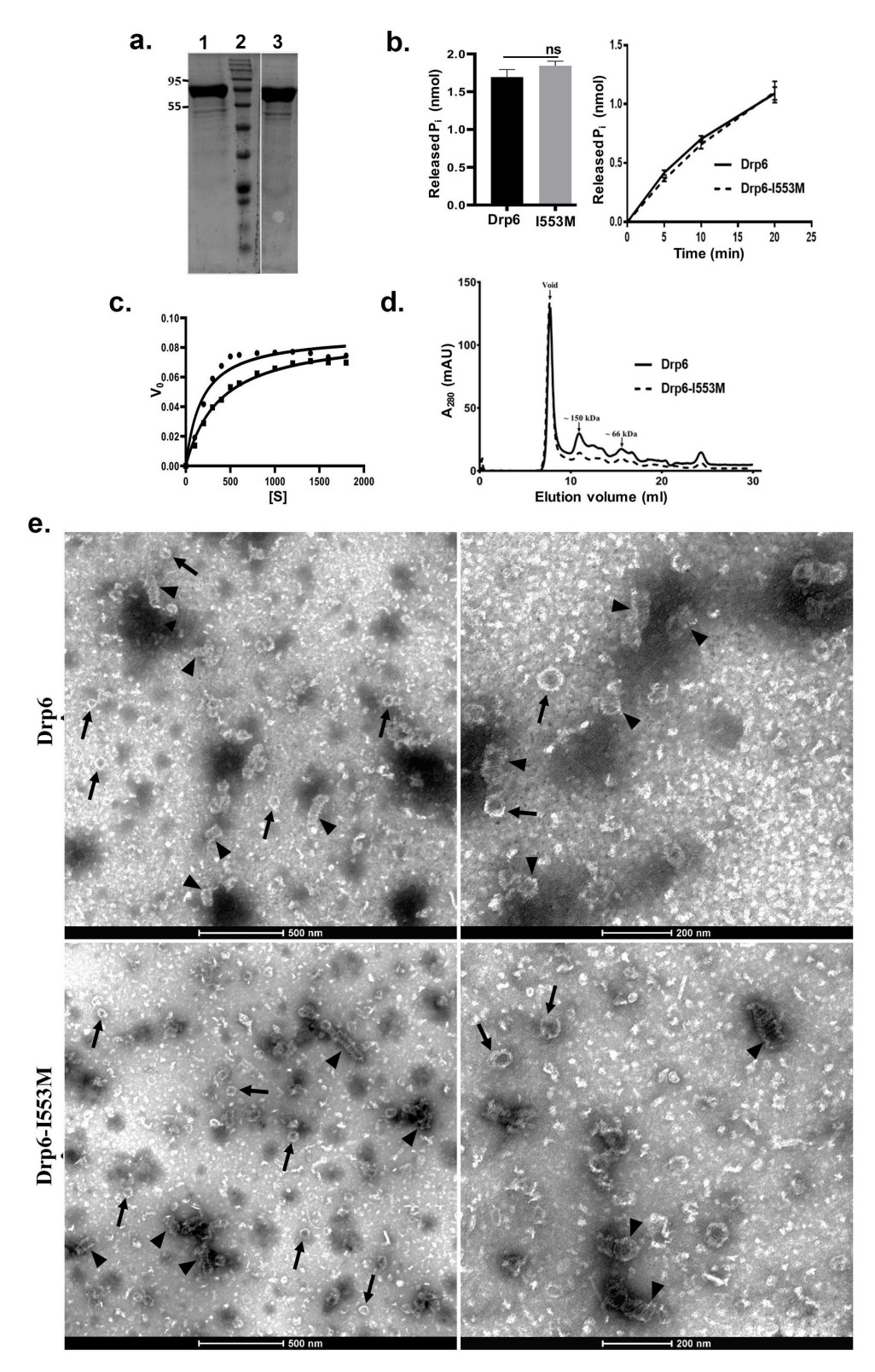

**Figure 4.** Mutation at I553 does not affect GTP hydrolysis activity and self-assembled structures. (a) Coomassie-stained SDS–PAGE gel showing purified His-drp6 (lane 1) and His-drp6-I553M (lane 3). Lane 2 is molecular weight marker. The positions of molecular weight are indicated on the left. (b) Graph showing GTP hydrolysis of His-drp6 (Drp6) and His-drp6-I553M (I553M) as measured by phosphate release after 30 min of reaction (left). The graph on right shows reactions carried out for 0–20 min. The statistical analysis was performed using unpaired t-test and the difference was non-significant

*Figure 4 continued on next page*

Figure 4 continued

(p≤0.0001). For both the experiments, n = 3. All the experiments were performed more than three times. (**c**) Michaelis–Menten plot showing GTP hydrolysis by His-drp6 (circle) and His-drp6-I553M (square). $V_0$ = rate of product formation in nmol $P_i$/μM protein/min and [S] = GTP concentration in μM. n = 3. (**d**) Chromatograms depicting elution profiles of His-drp6 and His-drp6-I553M using superdex 200 size exclusion column. The void volume and the positions of molecular weight markers are indicated by arrows. (**e**) Electron micrographs of negatively stained His-drp6 (Drp6) and His-drp6-I553M (Drp6-I553M) at two different magnifications. Helical spirals and the ring structures are found in both wild-type and mutant proteins and are indicated by arrow head and arrow, respectively.

The online version of this article includes the following source data for figure 4:

**Source data 1.** Mutation at I553 does not affect GTPase activity and self-assembly of Drp6.

higher order oligomeric structures (*Figure 6c*). A small fraction appearing as a mixture of monomers and oligomers with a peak around the dimer was also observed in the chromatogram. The GTPase assay results suggested that the I→A substitution does not affect its activity as assessed by the rate of GTP hydrolysis (*Figure 6d,e*). Based on these results, we conclude that the loss of Drp6-I553A nuclear recruitment is not due to the defect in GTPase activity or in the self-assembly of Drp6. The results also establish that Ile at the 553rd position is essential for Drp6-CL interactions and is a critical determinant of Drp6 nuclear recruitment.

We next examined whether the loss of interaction between the mutant proteins and CL is due to a change in the overall folding or conformation of Drp6. To assess overall folding, circular dichroism (CD) spectra of the mutants were compared with that of wild-type protein (*Figure 6f*). The analysis of far-UV CD spectra showed that Drp6 contains both α-helix and β-sheets as secondary structures with an ellipticity minima at 222 nm. Similar spectra were observed for I553M and I553A mutants. These results suggest that the mutant proteins were able to fold correctly. To rule out any change in the three-dimensional conformation due to the amino acid substitutions, we performed fluorescence spectroscopic analysis using intrinsic fluorescence of the lone tryptophan present in the Drp6 protein. The fluorescence spectra of the wild-type Drp6 showed a peak at around 332 nm, suggesting that the tryptophan residue is mostly buried (*Figure 6g*). The fluorescence spectra of I553M and I553A mutants did not show significant difference from that of wild type, suggesting that the overall structure of the proteins did not change due to mutations. To further assess the conformation around the I553 residue, acrylamide quenching experiments were performed using intrinsic tryptophan fluorescence. Since the single tryptophan (W548) is located only five amino acid residues apart from the I553, it allowed us to monitor the effect of mutations on the local conformation around this isoleucine residue. The accessibility of the tryptophan was measured by determining the quenching constant ($k_q$) for wild-type protein as well as the mutant proteins (*Figure 6h*). The results showed comparable values of $k_q$ (1.01 $M^{-1}$ $ns^{-1}$ for Drp6; 1.11 $M^{-1}$ $ns^{-1}$ for Drp6-I553M and 1.04 $M^{-1}$ $ns^{-1}$ for Drp6-I553A) for all the three proteins, suggesting that there is no major change in the local conformation due to mutations. These results lead us to conclude that the loss of CL interaction with I553A or I553M mutant is not due to a change in overall folding or local conformation of the mutant proteins. Taken together, it can be concluded that a single isoleucine residue in the membrane-binding domain is essential for interaction with CL and targeting Drp6 to the nuclear membrane.

## The E552 and M554 residues are not essential for CL binding and nuclear association of Drp6

We next investigated the role of two amino acid residues (E552 and M554) at the vicinity of I553 in CL binding and nuclear association. For this purpose, two independent mutants *drp6-E552D* and *drp6-M554L* were expressed either as N-terminal GFP-fusion proteins to assess their localizations in *Tetrahymena* cells or as N-terminal histidine fusion proteins for determining CL binding *in vitro*. Confocal microscopic analysis of the *Tetrahymena* cells expressing either *GFP-drp6-E552D* or *GFP-drp6-M554L* showed that, in addition to cytoplasmic puncta, both the mutant proteins also associated with nuclear envelope (*Figure 7a*). Floatation assay was performed to evaluate whether these two mutants also bind to CL-containing membrane *in vitro*. Both Drp6-E552D and Drp6-M554L appeared on the top fractions in the floatation experiments, suggesting their binding to CL (*Figure 7b*). Therefore, we conclude that the amino acid residues neighboring I553 do not determine the specificity for CL binding and nuclear recruitment of Drp6. Thus, the isoleucine at the 553rd position confers specificity for CL mediated nuclear recruitment of Drp6.

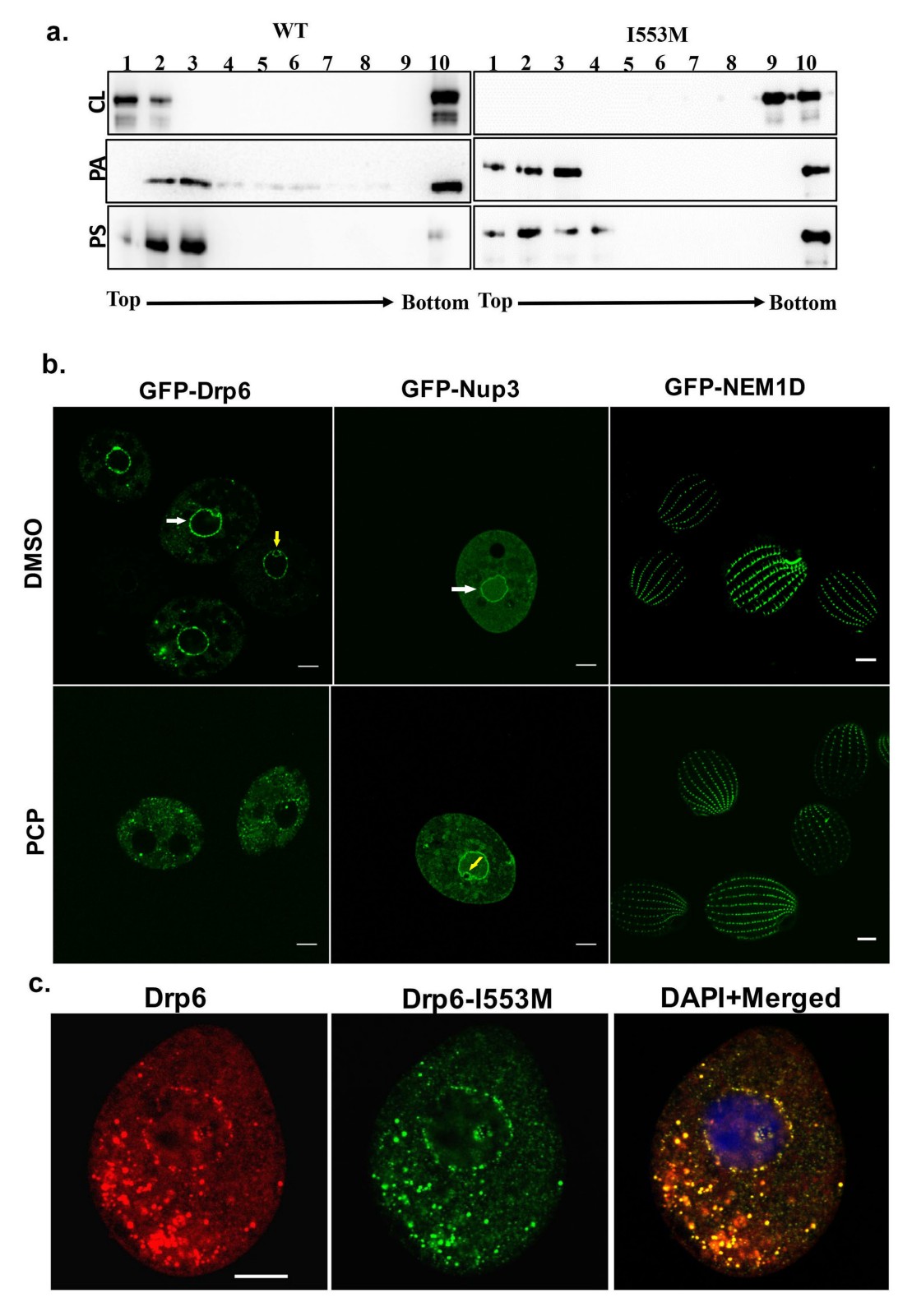

**Figure 5.** Interaction of CL with membrane-binding domain recruits Drp6 to the nuclear membrane. (a) Floatation assay was performed using liposomes with same composition and analyzed by western blotting as mentioned in *Figure 2c*. The assay was performed with His-drp6 (WT) and His-drp6-I553M (I553M). Liposomes supplemented with 10% CL, 10% PA, 10% PS were used for the assay. Fractions collected from top to bottom are indicated. Experiments were repeated at least three times and representative results are shown here. Mutation at I553 lost interaction completely with

*Figure 5 continued on next page*

*Figure 5 continued*

the liposomes containing CL while retaining interactions with liposomes containing either PS or PA, suggesting isoleucine residue at 553rd position is important for binding with CL in the bilayers. (**b**) Confocal images of live *Tetrahymena* cells expressing GFP-drp6 (left panel), GFP-Nup3/MacNup98B (middle panel), and GFP-Nem1D (right panel) either in presence (PCP) or absence (DMSO) of PCP. While localization on the nuclear envelope of MAC is indicated by white arrow, yellow arrow indicates MIC. Bar = 10 µm. (**c**) Confocal images of fixed *Tetrahymena* cells co-expressing mCherry-drp6 (left panel) and GFP-drp6-I553M (middle panel). Merged image with DAPI-stained nucleus is shown in right panel. Yellow color in the merged image signifies the presence of both Drp6 and Drp6-I553M in the same complex. Bar = 10 µm.

## Interaction of CL with Drp6 via membrane-binding domain is required for nuclear expansion

Previously, it has been shown that Drp6 is essential for MAC development in *Tetrahymena* (*Rahaman et al., 2008*). We have now established that interaction of Drp6 with CL is important for nuclear recruitment. Therefore, we hypothesized that inhibition of CL-Drp6 interaction would inhibit Drp6 function in MAC development. We took two independent approaches to perturb the interaction between CL and Drp6 and assessed the effect on MAC development. In the first approach, CL was depleted by treating cells at a stage prior to MAC development with PCP and then measuring the efficiency of new MAC formation in conjugating cells (*Figure 8a*). Quantitative analysis showed that while 71 ± 3.6% of the conjugants developed MACs in the control pairs, only 24 ± 1.7% developed MACs in the PCP-treated pairs (*Figure 8a*). In the second approach, we perturbed CL–Drp6 interaction by treating conjugants with nonyl acridine orange (NAO). NAO interacts with CL with very high affinity and has been used in mammalian cells to block interactions between CL and mitochondrial proteins involved in electron transport (*Maftah et al., 1990*). Exposing conjugants to NAO significantly inhibited new MAC development (*Figure 8a*).

In conjugating *Tetrahymena*, MAC development is not the only phenomenon requiring nuclear expansion. At a prior stage, the germline micronuclei (MICs) show dramatic elongation (*Cole et al., 1997*). We found that PCP treatment did not affect the frequency of elongation, that is, the percentage of pairs showing elongated MICs, but did produce a decrease in the extent of MIC elongation (*Figure 8a*). This inhibition of elongation had no detectable consequences for the subsequent stage of MIC meiosis. Treatment with NAO had no measurable effect on MIC elongation (*Figure 8a*) or the subsequent meiosis. These results are consistent with the idea that the key requirement for CL is during MAC expansion.

We next reasoned that if interaction of Drp6 with CL is important for MAC expansion, then over-expression of the isolated Drp6 membrane-binding domain might competitively inhibit the interaction and block Drp6 function during MAC expansion. To test this possibility, we expressed *GFP-drp6-DTD* in *Tetrahymena* and then allowed the cells to conjugate with a Drp6 wild-type strain. We measured MAC development in these pairs, and in pairs from a parallel WT x WT cross, at 8 hr during conjugation. In the control cross, more than 75% of pairs developed new MACs. In striking contrast, only 4–5% of pairs developed normal MACs in the pairs that included *GFP-drp6-DTD*-expressing cells (*Figure 8b*). Therefore, MAC development was almost completely blocked when *GFP-drp6-DTD*-expressing cells comprised one of the conjugation partners (*Figure 8b*).

In similar experiments, we also asked whether the expression of *GFP-drp6ΔDTD* might have a dominant-negative inhibitory effect on Drp6 function. Indeed, we found that in pairs where one cell over-expressed the ΔDTD construct, the pairs showed inhibition of MAC development that was similar to that induced by expression of the isolated DTD domain (*Figure 8b*). Neither the expression of *GFP-drp6-DTD* nor *GFP-drp6ΔDTD* significantly reduced the fraction of pairs showing elongated MICs (*Figure 8b*).

In conclusion, prior experiments with *DRP6* gene knockout pointed to a specific function in MAC development. Our current results from over-expression of mutant alleles are consistent with this idea. Taken together, our results support a model in which interaction of Drp6 with CL, for which a single isoleucine residue acts as a key determinant, is critical for nuclear targeting and therefore for MAC expansion.

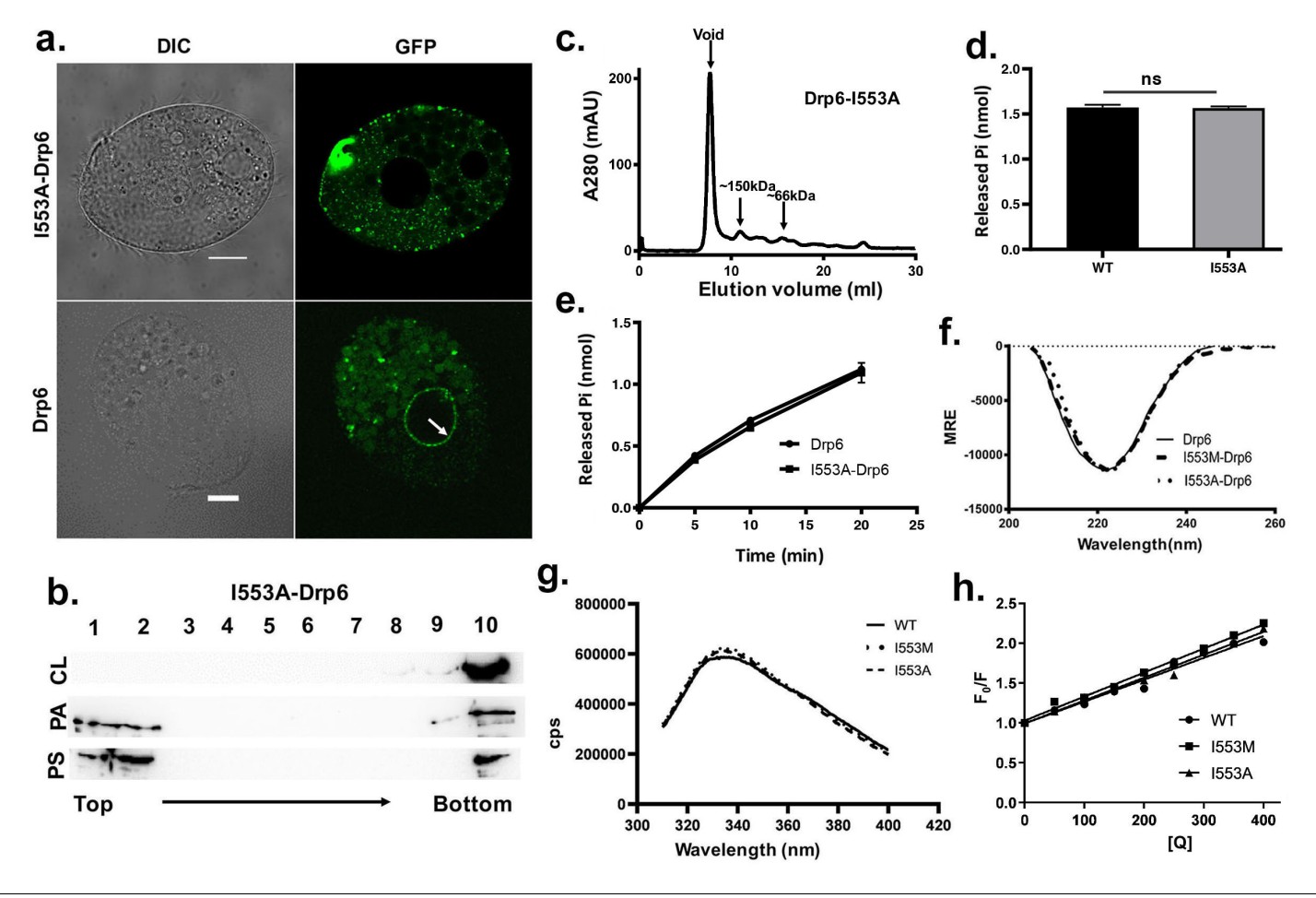

**Figure 6.** The loss of nuclear localization and CL binding by substitution mutations of I553 are not due to changes in overall structure and conformation. (a) Confocal images of live *Tetrahymena* cells expressing either GFP-drp6- I553A (top) or GFP-drp6 (bottom). Localization in the nuclear envelope is indicated by arrow. Bar = 10 μm. (b) Floatation assay showing binding of His-drp6-I553A with liposomes containing 10% CL (top), 10% PA (middle), and 10% PS (bottom) as analyzed by western blotting. Mutation of isoleucine to alanine at 553rd position results in loss of binding specifically with CL. (c) Elution profile of His-drp6-I553A in size exclusion chromatography. Similar to wild-type Drp6, I553A mutant also elutes mostly as higher order oligomeric structures. (d) GTP hydrolysis activities of His-drp6 (WT) and His-drp6-I553A (I553A) as measured after 30 min. The statistical analysis was performed using unpaired t-test and the difference was non-significant (p≤0.0001). n = 3. All the experiments were repeated several times. (e) Same as in (d) except the GTP hydrolysis was carried out 0–20 min. Drp6 = His-drp6 and I553A-Drp6 = His-drp6-I553A. (p≤0.0001). n = 3. All the experiments were repeated several times. (f) The graph showing the CD spectra of His-drp6, His-drp6-I553M, and His-drp6-I553A recorded from 205 nm to 260 nm. (g) Tryptophan fluorescence emission spectra of His-drp6 (WT), His-drp6-I553M (I553M), and His-drp6-I553A (I553A) with excitation at 295 nm. (h) Stern-Volmer plots of tryptophan fluorescence quenching by acrylamide. The emission at 332 nm (emission peak) was plotted.

The online version of this article includes the following source data and figure supplement(s) for figure 6:

**Source data 1.** Mutations at I553 results in loss of nuclear localization and cardiolipin interactions without affecting GTPase activity, self-assembly, overall folding and 3-D conformation of Drp6.

**Figure supplement 1.** Confocal images of fixed *Tetrahymena* cells after DAPI-staining either expressing GFP-drp6 (Drp6) or GFP-drp6-I553A (Drp6-I553A).

## Discussion

Drp6 is a nuclear dynamin and is involved in nuclear remodeling (*Rahaman et al., 2008*). In the present study, we have identified a membrane-binding domain in Drp6 that directly interacts with lipids. Inhibition of CL synthesis blocks nuclear localization of Drp6, suggesting that it plays a critical role in Drp6 recruitment. Further evidence of CL interaction determining nuclear localization of Drp6 comes from the mutation of isoleucine at the 553rd position. The mutant Drp6 proteins lose their nuclear localizations with concomitant loss of interaction with CL without affecting other properties such as

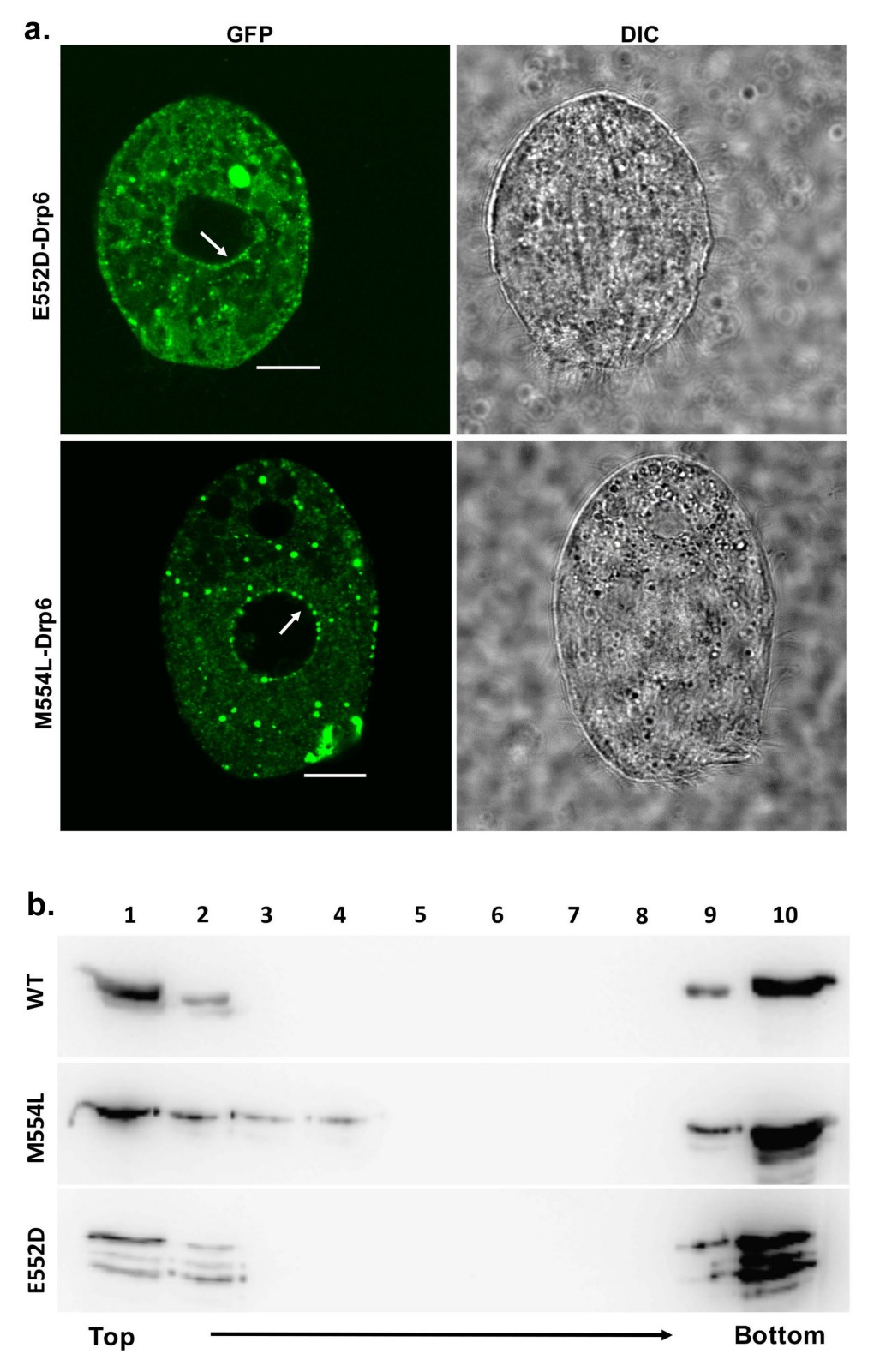

**Figure 7.** The mutations at E552 and M554 residues do not abrogate CL binding and nuclear localization. (a) Confocal images of live *Tetrahymena* cells expressing GFP-drp6-E552D (top) or GFP-drp6-M554L (bottom). Nuclear localization is indicated by arrow. Both E552D and M554L mutants associate with nuclear envelope. Bar = 10 μm. (b) Floatation assay was performed with liposomes containing 10% CL using His-drp6 (WT), His-drp6-M554L (M554L), and His-drp6-E552D (E552D).

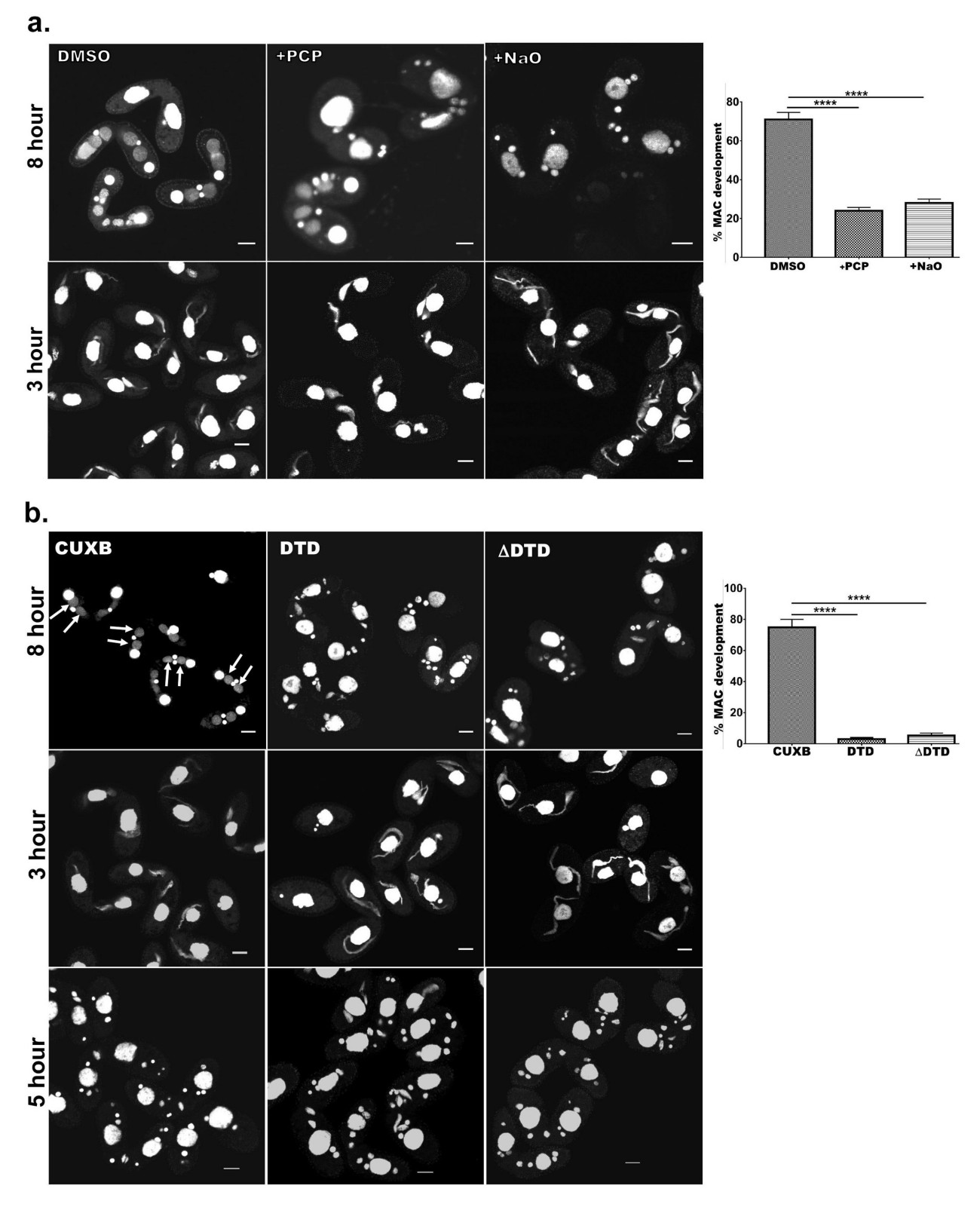

**Figure 8.** Cardiolipin and membrane-binding domain regulate macronuclear expansion. (a) Confocal images of fixed and DAPI-stained conjugation pairs of *Tetrahymena* at 8 hr and 3 hr post-conjugation. Two wild-type strains of *Tetrahymena* (Cu428 and B2086) were conjugated and treated either with pentachlorophenol (+PCP) or with nonyl acridine orange –D (+NaO) or with DMSO (DMSO). Top panel shows MAC development at 8 hr, and bottom panel shows MIC elongation at 3 hr. Percent MAC development at 8 hr is shown at the right. (b) Confocal images of fixed DAPI-stained

*Figure 8 continued on next page*

*Figure 8 continued*

*Tetrahymena* cells conjugated either between CU428 and B2086 (CUXB) or between CU428 and GFP-drp6-DTD-expressing cells (DTD) or between CU428 and GFP-drp6 ΔDTD-expressing cells (ΔDTD). Top panel MAC development stage at 8 hr, middle panel MIC elongation stage at 3 hr, and bottom panel meiotic stage at 5 hr. Percent MAC development at 8 hr is shown at the right. The newly developed MAC is indicated by arrow. For quantitation, three independent experiments were performed and analyzed by unpaired t-test (****p≤0.0001). n > 500.

The online version of this article includes the following source data for figure 8:

**Source data 1.** Inhibition of Drp6-CL interaction inhibits macronuclear development in *Tetrahyemena*.

GTPase activity and self-assembly into rings/helical spirals. The GFP-drp6-I553M recruited to the nuclear envelope only when co-expressed with wild-type mCherry-drp6, indicating that the mutant protein co-assembles with the wild type (*Figure 5c*). These results suggest that Drp6 molecules self-assemble on the nuclear envelope, and localizing the oligomer to the envelope does not require all subunits interact with CL. We further demonstrate that Ile at position 553 in the DTD is the sole residue critical for CL interactions, whereas the adjacent two amino acid residues (Glu at 552nd position and Met at 554th position) are not essential for CL interaction specificity. Taken together, these results suggest that a single isoleucine residue in the membrane-binding domain specifies the recruitment of Drp6 to the nuclear membrane.

We investigated the significance of CL in the nuclear remodeling function of Drp6. Inhibition of CL synthesis as well as perturbation of its interaction with Drp6 phenocopy the loss-of-function phenotype of Drp6, suggesting a role for CL in Drp6-mediated nuclear remodeling. Over-expression of *drp6-DTD* or *drp6ΔDTD* inhibits MAC expansion. Since DTD interacts with lipid including CL, the inhibition of MAC expansion is expected to be by competing with Drp6-CL interaction, concomitantly inhibiting Drp6 recruitment to the nuclear envelope. Inhibition of nuclear expansion by ΔDTD can be due to the inhibition of Drp6 localization on the nuclear envelope by forming a heterogenic complex as it lacks membrane-binding domain and does not associate with nuclear envelope. Based on these results, it can be concluded that CL acts as a molecular determinant in recruiting Drp6 on the nuclear envelope to perform nuclear remodeling function.

Drp6 is involved in nuclear expansion, which requires the incorporation of new lipids into the existing nuclear membrane, suggesting a membrane fusion function for Drp6. Consistent with this, we recently observed that Drp6 is able to perform membrane fusion *in vitro* (our unpublished results). Membrane fission or fusion involves exchange of lipids between two juxtaposed bilayers, and optimum membrane fluidity is likely to be essential for the exchange of lipids between adjacent leaflets. CL is known to facilitate the formation of apposed bilayers as well as to enhance membrane fluidity (*Unsay et al., 2013*). Dynamin proteins remodel membrane and bring bilayers to the vicinity during fission or fusion of membranes (*Praefcke and McMahon, 2004*). Therefore, interaction of Drp6 with CL may enhance bilayer interaction and membrane fluidity. Taking together, it is reasonable to conclude that Drp6-CL interaction on the nuclear envelope facilitates nuclear expansion by enhancing membrane fusion and hence is essential for macronuclear expansion.

Drp6 interacts with three different lipids namely CL, PA, and PS (present study). These interactions with multiple lipids might explain the localization of Drp6 in multiple sites (*Figure 9*). The target specificity is often determined by the interaction of the membrane-binding domain with the specific lipids on the membrane. Although DRPs including Drp6 lack PH domain, they harbor a membrane-binding domain at the corresponding location (*Figures 1a and 3a*; *Ramachandran and Schmid, 2018*). The sequence diversity in this domain might explain the diverse functions of the family members on different target membranes. While PIP2 present in the plasma membrane associates with endocytic dynamin at the neck of vesicles, CL exclusively present in the mitochondria is recognized by the dynamins possessing mitochondrial remodeling function (*Francy et al., 2017*; *Kameoka et al., 2018*). It is important to note that the nuclear envelope of *Tetrahymena* contains 3% CL (*Nozawa et al., 1973*), and Drp6 (which is specifically present in the ciliate *Tetrahymena*) has evolved to interact with CL for specific recruitment to the nuclear envelope. In addition to nuclear envelope, Drp6 is also associated with ER vesicles (*Rahaman et al., 2008*). Localization of Drp6-I553M/Drp6-I553A mutants on ER vesicles and their ability to interact with PS and PA on the membrane suggest that Drp6 localization on ER is dependent on either PA or PS or combination of both. Considering the abundance of PA on the ER membrane (*Pillai et al., 2017a*; *Pillai et al., 2017b*; *Zegarlińska et al., 2018*), it could be argued that PA is involved in recruitment of Drp6 to ER. We

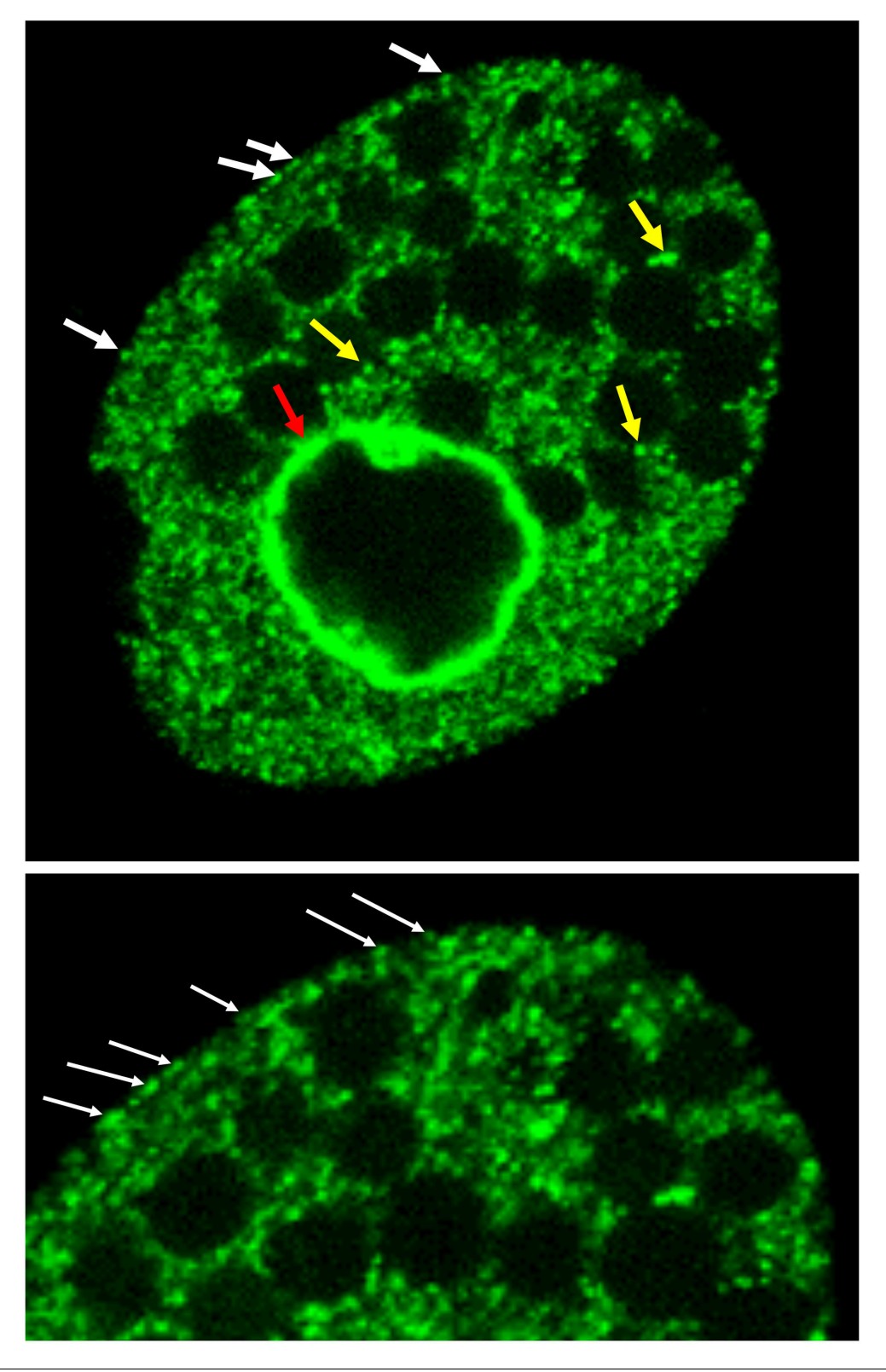

**Figure 9.** Drp6 localizes to multiple sites. Top: Confocal image of a live *Tetrahymena* cell expressing Drp6 as GFP-fusion protein. The localization of Drp6 on nuclear envelope (red arrow), on ER vesicles (yellow arrow), and on plasma membrane (white arrow) is shown. Bottom: Part of the image is enlarged to visualize the localization on the plasma membrane more distinctly.

also observed localization of Drp6 on plasma membrane (*Figure 9*), and since PS is also present in the plasma membrane (*Kay et al., 2012*), it is possible that Drp6 associates with plasma membrane via its interaction with PS. Although further experiments are required to find out the role of PS and PA in the recruitment of Drp6 in plasma membrane and ER, our results clearly show that lipid molecules play critical role in compartmentalizing the localization of Drp6 where CL shifts the dynamics from ER vesicles to nuclear envelope.

As mentioned earlier, I553 in the Drp6 membrane domain is critical for conferring specificity to Drp6 for recognizing CL on the nuclear envelope. The importance of I553 suggests the importance of hydrophobic patches for target membrane specificity, since CL is known to interact strongly with hydrophobic residues (*Planas-Iglesias et al., 2015*). This is substantiated in the endocytic dynamin that uses a hydrophobic region including isoleucine at the 533rd position in the PH domain for insertion into the target membrane (*Ramachandran et al., 2009*). Our results on isoleucine mutation within the membrane domain also suggest that the presence of a similar hydrophobic patch in Drp6 that is important for membrane interaction specifically via CL present on the nuclear envelope. However, hydrophobicity is not the sole determinant for the specific interaction with CL since methionine, which is also a strong hydrophobic residue, does not interact with CL when substituted for isoleucine (*Figure 5a*). Therefore, it is conceivable that, in addition to hydrophobicity, the side chain of isoleucine plays a critical role in conferring the specificity for the recruitment to the target membrane via interaction with CL. Although residues important for target membrane selection have been identified in many dynamin proteins, they involve a stretch of positively charged amino acids for recognizing anionic head groups of several lipids including CL, PS, and PIP2 (*Salim et al., 1996*; *Achiriloaie et al., 1999*; *Vallis et al., 1999*; *Rujiviphat et al., 2009*; *von der Malsburg et al., 2011*; *Bustillo-Zabalbeitia et al., 2014*; *Smaczynska-de Rooij et al., 2016*; *Wang et al., 2019*). However, it is not known how different dynamin proteins distinguish different lipids solely based on ionic interaction. In the present study, we demonstrate that a single isoleucine in the membrane-binding domain of Drp6 provides the specificity for CL. This isoleucine residue, however, does not influence Drp6 interaction with PS or PA, hence distinguishes among different negatively charged lipids. This is the first example of any dynamin protein in which a single amino acid site is shown to be critical for specific lipid interactions, thereby providing an additional target membrane (nuclear membrane) binding property to the protein. In conclusion, our results provide the underlying mechanism of target membrane selection by a nuclear dynamin and underscore the importance of CL interaction with a single amino acid residue in the Drp6 membrane-binding domain in facilitating nuclear expansion.

# Materials and methods

## Key resources table

| Reagent type (species) or resource | Designation | Source or reference | Identifiers | Additional information |
|---|---|---|---|---|
| Strain, strain background (*Tetrahymena thermophila*) | CU428.2 | Tetrahymena Stock Center, Cornell University, USA | RRID:TSC_SD00178 | Wild type |
| Strain, strain background (*Tetrahymena thermophila*) | B2086.2 | Tetrahymena Stock Center, Cornell University, USA | RRID:TSC_SD00709 | Wild type |
| Recombinant DNA reagent | pVGF (plasmid) | Meng-Chao Yao, FHCRC, Seattle, Washington | | rDNA-based Tetrahymena expression vector |
| Recombinant DNA reagent | GFP-drp6 (plasmid) | This study | | Drp6 cloned in pVGF expression vector |
| Recombinant DNA reagent | GFP-drp6-DTD (plasmid) | This study | | pVGF vector that expresses aa 517–600 of Drp6 |

*Continued on next page*

*Continued*

| Reagent type (species) or resource | Designation | Source or reference | Identifiers | Additional information |
|---|---|---|---|---|
| Recombinant DNA reagent | *GFP-drp6-ΔDTD* (plasmid) | This study | | pIGF vector expressing Drp6 with deletion of aa 517–600 |
| Recombinant DNA reagent | *GFP-drp6-I553M* (plasmid) | This study | | pIGF vector that expresses Drp6-I553M |
| Recombinant DNA reagent | *GFP-drp6-I553A* (plasmid) | This study | | pIGF vector that expresses Drp6-I553A |
| Recombinant DNA reagent | *GFP-drp6-E552D* (plasmid) | This study | | pIGF vector expressing Drp6-E552D |
| Recombinant DNA reagent | *GFP-drp6-M554L* (plasmid) | This study | | pIGF vector that expresses Drp6-M554L |
| Recombinant DNA reagent | *GFP-Nup3*/MacNup98B (plasmid) | Prof. D.L. Chalker, Washington University in St. Louis | | NCVB vector expressing GFP-Nup3. Blasticidine resistance |
| Recombinant DNA reagent | *GFP-Nem1D* (plasmid) | Previous study in lab (*Shukla et al., 2018*) | | pIGF vector expressing GFP-NEM1D |
| Recombinant DNA reagent | *mCherry-drp6* (plasmid) | This study | | NCVB vector that expresses mCherry-DRP6 |
| Recombinant DNA reagent | *His-drp6* (plasmid) | Previous study in the lab (*Kar et al., 2018*) | | pRSETB vector expressing Drp6 as N-terminal histidine tag |
| Recombinant DNA reagent | *His-drp6-DTD* (plasmid) | This study | | pRSETB vector expressing aa 517 to 600 as N-terminal histidine tag |
| Recombinant DNA reagent | *His-drp6-I553M* (plasmid) | This study | | pRSETB vector expressing Drp6-I553M as N-terminal histidine tag |
| Recombinant DNA reagent | *His-drp6-I553A* (plasmid) | This study | | pRSETB vector expressing Drp6-I553A as N-terminal histidine tag |
| Recombinant DNA reagent | *His-drp6-E552D* (plasmid) | This study | | pRSETB vector expressing Drp6-E552D as N-terminal histidine tag |
| Recombinant DNA reagent | *His-drp6-M554L* (plasmid) | This study | | pRSETB vector expressing Drp6-M554L as N-terminal histidine tag |
| Antibody | Anti-His6-peroxidase (mouse monoclonal) | Sigma–Aldrich | Cat#: 11965085001 | WB (1:5000) |
| Antibody | Anti-GFP (rabbit polyclonal) | Sigma–Aldrich | Cat#: AB10145 | WB (1:4000) |
| Antibody | Anti-Rabbit IgG-peroxidase (Goat polyclonal) | Sigma–Aldrich | Cat#: A0545-1ML | WB (1:80,000) |
| Chemical compound, drug | Pentachlorophenol | Sigma–Aldrich | Cat#: 87-86-5 | |

*Continued on next page*

*Continued*

| Reagent type (species) or resource | Designation | Source or reference | Identifiers | Additional information |
|---|---|---|---|---|
| Chemical compound, drug | Acridine orange 10-nonyl bromide | Invitrogen | Cat#: A1372 | |
| Chemical compound, drug | Phosphatidyl choline | Avanti Polar Lipids | Cat#: 840051 | |
| Chemical compound, drug | Phosphatidyl ethanolamine | Avanti Polar Lipids | Cat#: 840021 | |
| Chemical compound, drug | Cardiolipin | Avanti Polar Lipids | Cat#: 840012 | |
| Chemical compound, drug | Phosphatidic acid | Avanti Polar Lipids | Cat#: 840101 | |
| Chemical compound, drug | Phosphatidylserine | Avanti Polar Lipids | Cat#: 840032 | |
| Chemical assay or kit | BIOMOL green | Enzo Life Sciences | Cat#: BML-AK111 | |
| Chemical compound | uranyl acetate | MP Biomedicals | Cat#: 181561 | |
| Chemical compound | Ni-NTA agarose | QIAGEN | Cat#: 30210 | |
| Other | Membrane lipid strips | Echelon Biosciences | Cat#: P-6002 | |
| Other | Carbon film 200 mesh copper | Ted Pella Inc | Cat#: CF200-CU | |
| Other | DAPI stain | Invitrogen | Cat#: D1306 | (0.25 µg/ml) |

## *Tetrahymena* strains and culture conditions

*Tetrahymena thermophila* CU428 and B2086 strains were obtained from Tetrahymena Stock Center (Cornell University, USA). Cells were cultured in SPP medium (2% proteose peptone [BD, USA], 0.2% glucose, 0.1% yeast extract, and 0.003% ferric EDTA) at 30°C under shaking at 90 rpm. For conjugation, mating type cells were grown in SPP media to a density of $3 \times 10^5$ cells/ml, washed and resuspended in DMC media (0.17 mM sodium citrate, 0.1 mM $NaH_2PO_4$, 0.1 mM $Na_2HPO_4$, 0.65 mM $CaCl_2$, 0.1 mM $MgCl_2$), and incubated at 30°C, 90 rpm for 16–18 hr. To initiate conjugation, starved cells of two different mating types were mixed and incubated at 30°C without shaking. All the reagents were purchased from Sigma–Aldrich unless mentioned otherwise.

## Cloning and expression of transgenes in *Tetrahymena*

The Drp6ΔDTD lacking aa 517–600 was created by overlap PCR using a set of four oligonucleotides in a two-step PCR method. Drp6-E552D, Drp6-I553M, Drp6-I553A, and Drp6-M554L were generated by site-directed mutagenesis using Quick Change protocol (Stratagene). For expression in *Tetrahymena*, rDNA-based vector pVGF or pIGF was used. While the PCR products of Drp6 (aa 1–710) and Drp6-DTD (aa 517–600) were inserted between XhoI and ApaI restriction sites of pVGF, the Drp6-E552D, Drp6-I553M, Drp6-I553A, Drp6-M554L, and Drp6ΔDTD were introduced into pIGF using Gateway cloning strategy (Invitrogen) using the manufacturer's protocol and were expressed as N-Ter GFP-tagged fusion proteins. All the constructs were confirmed by sequencing. Conjugating wild-type *Tetrahymena* cells were transformed with these constructs by electroporation, and the transformants were selected using 100 µg/ml paromomycin sulfate. For the co-expression studies, *mCherry-drp6* was generated by introducing mCherry sequences between PmeI and XhoI sites followed by Drp6 sequences between XhoI and ApaI sites of NCVB vector. Co-transformants were

generated by biolistic transformation of linearized mCherry-drp6 nucleotide sequences into the cells expressing GFP-drp6-I553M and were selected in the presence of 60 µg/ml blasticidine and 120 µg/ml paromomycin sulfate supplemented with 1 µg/ml cadmium chloride.

Cells were grown to a log phase ($2.5–3.5 \times 10^5$ cells/ml), and expression was induced by adding cadmium chloride at concentration of 1 µg/ml for 4 hr. Cells were harvested at 1100 g, fixed with 4% paraformaldehyde for 20 min at RT, washed and resuspended in 10 mM HEPES, pH 7.5, and DAPI stained before imaging.

## Cloning, expression, and purification of recombinant proteins in *Escherichia coli*

For expression in *Escherichia coli*, the amplified PCR products of Drp6 and Drp6-DTD were cloned into pRSETB using BamHI and EcoRI sites. The Drp6-E552D, Drp6-I553M, Drp6-I553A, and Drp6-M554L were cloned in pRSETB vector using Quick Change protocol (Stratagene). The resulting constructs were transformed into chemically competent *E. coli* C41(DE3) cells and transformants were inoculated into LB broth supplemented with 100 µg/ml ampicillin and grown at 37°C till the $OD_{600}$ reached 0.4. The cultures were then shifted to 18°C and expression was induced after 1 hr by adding 0.5 mM IPTG (Sigma) and kept for 16 hours at the same temperature before harvesting the cells. The harvested cells were resuspended inice-cold buffer A (25 mM HEPES pH 7.5, 300 mM NaCl, 2 mM MgCl₂, 2 mM β-mercaptoethanol, and 10% glycerol) supplemented with EDTA-free protease inhibitor cocktail (Roche) and 100 mM phenyl methyl sulfonyl fluoride, lysed by sonication, and the lysates were centrifuged at 52,000 g for 45 min at 4°C. The supernatants were incubated with Ni-NTA agarose resin (Qiagen, Germany) for 2 hr before washing with 100 bed volume buffer A supplemented with 50 mM imidazole. The bound proteins were eluted with 250 mM imidazole in buffer A. The purified proteins were checked by Coomassie-stained SDS–PAGE gels, and the purity was assessed by Image J analysis (NIH, USA). The fractions containing the purified proteins were pooled, dialyzed with buffer A, and concentrated using Amicon ultra-15 filters (Merck-Millipore, Germany). Protein concentration was estimated by Bradford assay (Bio-Rad Laboratories, USA).

## Western blotting

Samples were subjected to SDS–PAGE gel, and the proteins were transferred to PVDF membrane. Membrane was blocked with 2% bovine serum albumin (BSA) in TBST (50 mM Tris–Cl, 150 mM NaCl, pH 8.0%, and 0.05% Tween 20) for 1 hr. The blot was then incubated with HRP-conjugated anti-His monoclonal antibody (1:5000) and detected with supersignalfemto substrate (Thermoscientific, USA) using ChemiDoc imaging system (Bio-Rad Laboratories, USA).

## Fractionation of membrane protein and soluble protein

*Tetrahymena* cells expressing GFP-drp6 or GFP-DTD were lysed in 500 µl of ice-cold lysis buffer containing 25 mM Tris–Cl pH 7.5, 300 mM NaCl, 10% glycerol supplemented with E-64, pepstatin, aprotinin, PMSF, and protease inhibitor cocktail (Roche) by passing through a ball-bearing homogenizer with a nominal clearance of 0.0007 in. The resulting lysates were centrifuged at 16,000 g for 15 min at 4°C, the supernatant was collected as soluble protein fraction, and the pellet containing the membrane fraction was resuspended in 500 µl lysis buffer. The proteins in both soluble fraction and membrane fraction were separated in 12% SDS–PAGE gel and analyzed by western blotting using anti-GFP polyclonal antibody (1:4000; Sigma–Aldrich).

## Lipid overlay assay

Total *Tetrahymena* lipid was extracted from growing *Tetrahymena* cells ($5 \times 10^5$ cells/ml) following the method by *Bligh and Dyer, 1959*. Drops of 5 µl in chloroform were spotted on the nitrocellulose membrane and incubated with His-drp6 (90 µg/ml) in GTPase assay buffer in the presence or absence of 1 mM GTP for 1 hr. In control experiments, BSA was used in place of His-drp6. The assay using membrane lipid strips (P-6002, Echelon Biosciences, USA) spotted with 100 pmol of 15 different lipids were used according to the manufacturer's instruction. The binding of proteins was detected by western blot analysis using anti-His monoclonal antibody.

## Floatation assay

Lipids (Avanti Polar) were dissolved in analytical grade chloroform, and liposomes were prepared using 2.5 mg total lipid in 1 ml chloroform. The liposomes contained 70% PC and 20% PE along with either 10% CL or 10% PA or 10% PS. A thin dry film was obtained by drying the solution in a round bottom flask, and solvent was completely removed in a lyophilizer. Liposomes were made by rehydrating the film in buffer A (25 mM HEPES pH 7.5, 2 mM MgCl$_2$, 150 mM NaCl) pre-warmed at 37°C. The resuspended solution was extruded 17–21 times through extruder (Avanti Polar) using filter with 100 nm pore, and the size distribution was measured by DLS in Malvern Zetasizer Nano. For floatation assay, 1 µM protein was incubated with 0.5 mg liposomes in buffer A supplemented with 1 mM GTP for 1 hr at RT. Sucrose was added to the reaction mixture (final sucrose concentration 40%), placed at the bottom of a 13 ml ultra-centrifugation tube, and overlaid with 2 ml each of 35%, 30%, 25%, 20%, 15%, and 0% sucrose solutions in the same buffer. The gradient was subjected to ultra-centrifugation in Beckman Coulter ultra-centrifuge at 35,000 RPM for 15 hr at 4°C using SW41 rotor. Fractions (1 ml each) were collected from top and detected by western blotting using anti-His HRP-conjugated monoclonal antibody (1:5000).

## Measurement of GTP hydrolysis activity

The GTP hydrolysis activity of recombinant Drp6, Drp6 I553M, and Drp6-I553A was measured in a colorimetric assay using Malachite Green-based phosphate assay reagent (BIOMOL Green, Enzo Life Sciences, USA). The GTPase assay (20 µl in 25 mM HEPES pH7.5, 15 mM KCl, 2 mM MgCl$_2$) was performed in the presence of 1 mM GTP (Sigma) using 1 µM protein for 0–30 min at 37°C. The reaction was stopped by adding 5 µl of 0.5 M EDTA, and absorbance was measured at 620 nm. For measuring K$_M$ and K$_{cat}$, reactions were performed for 10 min at 37°C in triplicate using varying concentrations of GTP (50 µM–2000 µM). The values obtained from three independent experiments were plotted and analyzed using GraphPad Prism7 software. The statistical analysis was performed by using unpaired t-test.

## Size exclusion chromatography

Size exclusion chromatography was performed on the Superdex 200 GL 10/300 column (GE Life Sciences, USA) using Akta Explorer FPLC system (GE Healthcare, USA), which was calibrated with standard molecular weight markers (Sigma–Aldrich, USA). Five hundred microliters of protein (0.5 mg/ml) in buffer A was loaded onto the pre-equilibrated column and was run at 0.5 ml/min. The chromatogram was recorded by taking absorbance at 280 nm.

## Electron microscopy

Purified recombinant Drp6 or Drp6 I553M (1 µM) was incubated with 0.5 mM GTPγS in 25 mM HEPES pH 7.5, 150 mM NaCl, and 2 mM MgCl$_2$ for 20 min at room temperature and was adsorbed for 2 min onto a 200 mesh carbon coated Copper grid (Ted Pella, Inc, USA). The grid was stained with a drop of 2% freshly prepared uranyl acetate (MP Biomedicals, USA) for 2 min and dried at room temperature for 10 min. The electron micrographs were collected on a FEI Tecnai G2 120 kV electron microscope.

## Confocal microscopy

*Tetrahymena* cells were fixed with 4% paraformaldehyde (PFA) in 50 mM HEPES pH 7.5 for 20 min at RT and were collected in 10 mM HEPES pH 7.5 after centrifugation at 1100 × g. The fixed cells were stained with DAPI (0.25 ug/ml) (Invitrogen, USA) and washed with 10 mM HEPES pH 7.5 before imaging. The images were collected in a Zeiss LSM780 or Leica DMI8 confocal microscope.

## Circular dichroism

CD measurements were recorded at 145 µg/ml protein in 25 mM HEPES pH 7.5, 300 mM NaCl, 2 mM MgCl$_2$, 2 mM β-mercaptoethanol using a Jasco J-1500 instrument (Jasco Inc, USA). Ellipticity (millidegrees) was measured between 260 and 205 nm. Mean residue ellipticity ([θ]$_{MRW}$ in deg cm$^2$ dmol$^{-1}$) for each wavelength was calculated by using the following formula:

$$[\theta]_{MRW} = \theta.100.M_r.(c.l.N_A)^{-1}$$

where θ is the ellipticity in mdeg, $M_r$ is the molecular weight of the protein in Da, c is protein concentration in mg/ml, l is the path length in centimeter, and $N_A$ is the number of amino acid residues of the protein.

## Fluorescence quenching assay

For the fluorescence spectroscopy experiments, 300 µl of 0.2 µM protein in 25 mM HEPES pH 7.5, 300 mM NaCl, 2 mM $MgCl_2$, and 2 mM β-mercaptoethanol were measured by excitation at 295 nm (2 nm band pass) in a 10 mm quartz cuvette using FLS-1000 (Edinburgh Instruments Ltd., UK). For the fluorescence quenching experiments, different acrylamide concentrations (0–400 mM) were added to 300 µl of 1.5 µM protein sample, and the spectra were recorded in the Cary Eclipse fluorescence spectrophotometer (Agilent Technologies, USA). The emission spectra were collected from 310 to 410 nm (10 nm band pass), and the values at 332 nm (the emission peak) were analyzed using *Stern–Volmer* equation, expressed as follows:

$$F_0/F = 1 + K_{sv}[Q]$$

where $F_0$ is the fluorescence in the absence of quencher and F is the fluorescence in the presence of quencher. Q denotes the concentration of the quencher in M. $K_{sv}$ is the *Stern–Volmer constant*. The obtained values were plotted using GraphPad Prism 7, and $K_{sv}$ was calculated from the slope of the graph. The quenching constant was calculated using the equation:

$$k_q = K_{sv}/\tau$$

Where $k_q$ is the quenching constant (in $Mol^{-1} ns^{-1}$) and τ is the life time (in ns) of tryptophan. The value of τ is considered to be 2.7 ns as reported earlier (*Ronda et al., 2018*).

## PCP and NAO treatment

The growing *Tetrahymena* cells either expressing *GFP-drp6* or *GFP-Nup3/MacNup98B* (a kind gift from Prof. Douglas Chalker, Washington University in St. Louis; *Malone et al., 2008*; *Iwamoto et al., 2009*) or *GFP-Nem1D* were treated with 10 µM PCP in DMSO for 30 min before fixing with 4% paraformaldehyde. The conjugation pairs were treated with either 30 µM PCP or 0.5 µM NAO (Invitrogen, USA) after 2.5 hr, 4.5 hr, or 7.5 hr post-mixing and were fixed after 30 min of addition. The cells were stained with DAPI (0.25 µg/ml) before imaging. The statistical analysis was performed by using unpaired t-test.

## Acknowledgements

We thank Prof. Aaron Turkewitz from the University of Chicago, Dr Kausik Chakraborty from IGIB, Prof. Jacek Gaertig from University of Georgia, Dr Utpal Nath from Indian Institute of Science, and Dr Renjith Mathew from National Institute of Science Education and Research for critical evaluation and useful comments on the manuscript. For other help, we thank members of this laboratory Sakti Ranjan Rout and Soham Mukhopadhyay. The work was partly funded by DBT grant (BT/PR14643/BRB/10/862/2010) to AR. The funders have no role in study design and interpretation, or the decision to submit the work for publication.

## Additional information

### Funding

| Funder | Grant reference number | Author |
| --- | --- | --- |
| Department of Biotechnology , Ministry of Science and Technology | BT/PR14643/BRB/10/862/2010 | Abdur Rahaman |

The funders had no role in study design, data collection and interpretation, or the decision to submit the work for publication.

## Author contributions
Usha Pallabi Kar, Himani Dey, Data curation, Validation, Investigation, Methodology; Abdur Rahaman, Conceptualization, Formal analysis, Supervision, Funding acquisition, Validation, Investigation, Writing - original draft, Writing - review and editing

## Author ORCIDs
Abdur Rahaman [ID] https://orcid.org/0000-0002-8440-9633

## Decision letter and Author response
Decision letter https://doi.org/10.7554/eLife.64416.sa1
Author response https://doi.org/10.7554/eLife.64416.sa2

## Additional files

### Supplementary files
- Transparent reporting form

### Data availability
All data generated or analysed during this study are included in the manuscript and supporting files.

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
