## [Decision Letter]

[Editors' note: this paper was reviewed by Review Commons.]

**Decision letter after peer review:**

Thank you for submitting your article "Cardiolipin targets a dynamin related protein to the nuclear membrane" for consideration by *eLife*. Your article has been reviewed by two external experts and the evaluation has been overseen by a Vivek Malhotra as the Senior Editor.

The reviewers have discussed the reviews with one another and the Reviewing Editor has drafted this decision to help you prepare a revised submission.

Summary:

The paper reports the involvement of isoleucine 553 in targeting Drp6 to cardiolipin containing nuclear membrane. The data are interesting, but there is no mechanistic understanding of how a single amino acid can target this protein so specifically to cardiolipin enriched membranes

Essential revisions:

The authors are strongly requested to address the issues that were raised in the previous review. The authors state in their rebuttal that they plan to address them in a timely manner.

The additional request of one reviewer that should be addressed is to test the involvement of residue 552 and 554 to highlight the significance of isoleucine in position 553 in targeting Drp6 to cardiolipin.

---

## [Author Response]

We thank all the reviewers for their critical evaluation and excellent suggestions to improve the manuscript. We have made all the changes suggested by the reviewers and performed the experiments to address the various concerns raised. Please find below our response to the reviewers’ comments:

Reviewer #1:1) This is an interesting study from the Rahaman group that identifies cardiolipin (CL) as a potential binding target for Drp6 recruitment to the nuclear membrane in Tetrahymena (that has a unique nuclear remodeling program). In addition, they identify a residue, I553 in the DTD region, which they claim is a key residue involved in specific CL interactions. While the experiments themselves are technically sound, and are well performed and controlled, I don't find the major conclusion that I553 is involved in direct CL interactions justified or well rationalized. By their own admission (in the Discussion), the conservative mutation I553M may perturb local folding and may indirectly affect CL interactions. There is no test of DTD folding with and without the I553M mutation, nor are there other mutations (e.g. I553A and in the vicinity) tested. CD experiments in the absence and presence of CL^-^containing membranes will likely yield information on the impact of the I553 mutations, while DLS experiments would inform on the hydrodynamic properties (overall 3D fold) of the DTD and the impact of these mutations. CL interactions generally involve a combination of electrostatic and hydrophobic forces. Where do the electrostatic interactions come from? Why would an Isoleucine to Methionine mutation affect the hydrophobic component, even if I553 is the key hydrophobic residue?

We thank the reviewer for appreciating our work. We agree that the exact mechanism of how I553 provides specificity to cardiolipin binding was not addressed adequately in the previous manuscript. We have now performed several experiments to address this issue. Our results clearly show that isoleucine at 553 position is critical for cardiolipin binding and nuclear recruitment of Drp6 and the interaction is direct.

To show that the interaction is direct we have generated another mutant I553A and performed all the experiments including nuclear localization, cardiolipin binding, self-assembly, GTPase activity. Similar to I553M, I553A mutant also failed to bind cardiolipin and to associate with nuclear membrane without affecting GTPase activity and self-assembly of Drp6.These results confirm that isoleucine is critical at this position. We have included these results in Figure 6A-E and Figure 6—figure supplement 1 in the revised manuscript.

To demonstrate that overall folding is not changed due to mutations at this position, we have performed CD spectroscopic analysis and found that mutations didn’t affect overall folding. We have also included these results in Figure 6F in the revised manuscript.

Additionally, we have performed fluorescence spectroscopy using the only tryptophan (W548) residue present nearby I553. The tryptophan fluorescence emission spectra of mutants (I553M and I553A) were similar to that of wildtype Drp6 suggesting there is no major change in conformation due to mutations. To evaluate any minor change in the local conformation due to mutations, we have performed acrylamide quenching experiments using the intrinsic tryptophan fluorescence. The results show that quenching constants of both the mutants were similar to that of wildtype Drp6, and therefore we conclude that the local conformation is not changed due to mutations. These results are included in Figure 6G and 6H in the revised manuscript.

Electrostatic interactions are important for all the phospholipids while interacting with protein and is expected to come from other amino acid residues which are positively charged. Electrostatic interaction may contribute to the affinity of the interaction by providing additional binding energy. But considering its universal nature of interaction with all the phospholipids, it cannot give specificity for a specific lipid and hence would not discriminate among different phospholipids.

Regarding the hydrophobic component, the reviewer is correct that both are strong hydrophobic amino acids and loss of I553M interaction with cardiolipin may not be due to change in hydrophobicity. This is now explicitly mentioned in the manuscript. Therefore, it appears that the side chain of isoleucine at 553 position is important for binding to cardiolipin.

Reviewer #1 (Significance (Required)):2) The addressed phenomenon is restricted to Tetrahymena and may not have far reaching implications. Regardless, the identification of CL as a binding target for Drp6 at the nuclear membrane of this organism is in itself significant. The conclusion that I553 is the key CL binding residue is however not warranted. Additional experiments are needed to dissect how this residue impacts CL interactions and examine whether the observed effect is direct or indirect.

We thank the reviewer for appreciating the significance of this work. We agree that our data is *Tetrahymena* specific. However, we believe that the study is potentially relevant for all the proteins whose association with target membranes depend on cardiolipin including many cardiolipin interacting DRPs (such as DRPs involved in biogenesis and maintenance of mitochondria).

As mentioned in the previous section, all the recent results now reinforce that the isoleucine is critical for binding cardiolipin and the interaction is direct.

3) The writing is not clear in some parts and may require a round of language editing. There are no issues with reproducibility.

We have rewritten the manuscript to the best of our abilities and corrected the language.

Reviewer #2:Reviewer #2 (Evidence, reproducibility and clarity (Required)):Dynamin is a GTPase superfamily protein involved in membrane fusion and division. This paper focused on Drp6, one of the eight dynamin superfamily proteins of Tetrahymena, and analyzed its nuclear envelope localization mechanism by a combination of in vivo cytogenetical analysis and in vitro biochemical analysis for the various mutant Drp6 proteins. Results showed that a specific amino acid residue (isoleucine at the 553rd) in the membrane binding domain of Drp6 was required for its nuclear membrane localization, but this residue is not required for ER/endosome localization and GTPase activity. Furthermore, in vitro floating analysis using centrifugation indicated that Drp6 specifically bound to the cardiolipin at the 553rd isoleucine residue and this binding was required for Drp6's nuclear membrane localization. Finally, removal of cardiolipin from the conjugating cells using inhibitor treatment showed that cardiolipin was required for the new macronucleus formation (including the expansion of macronuclear envelope) through the function of Drp6. Based on these results, authors concluded that cardiolipin targets Drp6 to the nuclear membrane in Tetrahymena.Major comments:The experimental data presented in this paper are reasonable and the results are solid, and therefore I think the deduced conclusions are convincing. However, to improve this paper, I have several minor comments to be revised before publication.Minor comments:1) In the previous paper, it has been shown that GFP-Drp6 is localized in the inner nuclear membrane of both macronucleus and micronucleus. In this paper, however, this point is not clearly stated and is not shown in the figures. I could not understand such localization pattern of GFP-Drp6 in Figure 1C and Figure 3B and the statements in the text. I suggest adding such statements somewhere in Introduction or Result section. Also, add adequate references to the corresponding statements in the text.

We thank the reviewer for pointing out this important aspect of localization. We have included the statement in the Introduction section with appropriate reference.

2) Related to the comment 1, I suggest replacing Figure 1C (images of fixed cells) with Figure 1—figure supplement 1B (images of live cells) because nuclear localization of GFP-Drp6 are much clearer in Figure 1—figure supplement S1B (live cell) than Figure 1C (fixed cell), and because fixation may cause artificial redistribution of the proteins. Please add arrows in those figures to point out the position of micronucleus in those figures if necessary.

The reviewer is correct about artificial redistribution of proteins that may arise from fixation. We thank the reviewer for the suggestion of replacing the fixed images with the live cell images in the main figure. We have now included live cell images in the main figure, and fixed cell images in the Figure 1—figure supplement 1 is also included to visualize the DAPI stained nucleus.

3) Similarly, I suggest replacing images of Figure 5B (fixed cells) with those of Figure 3—figure supplement (live cells).

We have changed accordingly as suggested by the reviewer.

4) GFP-Nup3 is used as a marker protein of the nuclear pore complex (NPC). However, there is no description of how GFP-Nup3 is obtained or made. Add description how this DNA plasmid was obtained or generated.

We have now incorporated the information about how GFP-Nup3 plasmid DNA was obtained.

5) Related to the comment 4, "Nup3" is first discovered in Malone et al., 2009, but also soon after discovered as the name of "MicNup98B" in Iwamoto et al., 2009 and used in several papers including Iwamoto et al., Genes Cells, 2010; JCS, 2015; JCS 2017; and more. Because Nup3 is the Tetrahymena paralogs of human Nup98 and the name of "Nup98" is well established to call these homologs in various eukaryotes, I suggest adding the name of "MacNup98B" after the word of "Nup3" for reader's better understanding. I also suggest adding appropriate references to refer to this protein as follows: Add Malone et al., 2009 for "Nup3" and Iwamoto et al., 2009 for "MacNup98B."

We thank reviewer for pointing out and we now incorporated the name "MacNup98B" after the word "Nup3". We have also included the references suggested.

6) page 9, line 295: I wonder if "Figure 3B" may be a mistake of "Figure 5C." If so, please correct this.

We thank the reviewer for noticing this mistake. We have now corrected the figure number accordingly.

7) page 10, the second paragraph (lines 311-322): This paragraph discussed the possible involvement of Drp6 in the nuclear envelope expansion of the post-zygotic nucleus. It may be interesting to point out that large-scale nuclear envelope reorganization including the formation of the redundant nuclear envelope and the type-switching of the NPC (from the MIC-type NPC to the MAC-type one) has been reported at this developmental stage (Iwamoto et al., JCS 2015). For example, the peculiar shaped nuclear envelope with the redundant/overlapping nuclear envelope structure can be seen and the MAC-type NPCs rapidly assembles to the expanding nuclear envelope. It may be interesting to point out that cardiolipin and Drp6 may be involved in these phenomena. But it is too speculative and therefore consider adding such a discussion as an option.

We thank the reviewer for pointing out the correlation between cardiolipin and Drp6 with type switching of the NPC. As also mentioned by the reviewer, the correlation is highly speculative. Therefore, we have not added this in the Discussion.

8) Is the word "GFP-drp6-I553M" written in italics intended for the gene for the GFP-drp6-I553M protein? If so, protein may be acceptable here. Make sure there are no problems with italicized characters. Also, check if the lowercase letter "d" in "drp6" is OK because large letters are used in other cases.

As suggested by the reviewer, we have now changed "*GFP-drp6-I553M*" to “GFP-Drp6-I553M”.

9) Figure 1: I recommend switching the positions of HDyn1 and Drp6 in Figure 1A to keep the order in Figure 1B.

We have now switched the positions of HDyn1 and Drp6 in Figure 1A.

10) page 21, line 671: Add the word "Tetrahymena" before "Drp 6" to pair with the word "human dynamin 1".

Suggested change is incorporated

11) page 23, line 729: Remove "and."

Suggested change is incorporated

12) page 23, lines 729 and 731: Unify the expression of "cardiolipin" and "Cardiolipin"

Suggested change is incorporated

13) page 23, line 732: Add "or" before "10% Phosphatidylserin."

Suggested change is incorporated

14) Figure 3A: Please mark the position of I553M in the figure if possible. Alternatively, indicate the range of amino acid residues after the words "red" and "green" in the figure legend.

As suggested by the reviewer, we have now incorporated the positions of the amino acid residues in the figure legend

We thank the reviewer for the excellent comments that “the experimental data presented in this paper are reasonable and the results are solid, and therefore I think the deduced conclusions are convincing.” We also thank the reviewer for the minor comments which are thorough and very insightful. it improved the manuscript substantially. We have incorporated all the changes in the revised manuscript.

Reviewer #2 (Significance (Required)):The corresponding author and his colleagues have reported that Tetrahymena Drp6 is localized to the outer nuclear membrane of both macronucleus and micronucleus of Tetrahymena (Elde et al., 2005) and that Drp6 is required for the formation of new macronuclei during nuclear differentiation (Rahaman et al., 2008). Therefore, these parts are not novel.The novelty of this study is as follows:1) The discovery of a specific amino acid residue (isoleucine at the 553rd) of Drp6 that is required for its nuclear membrane localization.2) the discovery of a lipid molecule, cardiolipin, as a critical partner for Drp6's nuclear membrane targeting.3) Discovery of involvement of cardiolipin in the new macronucleus formation (the expansion of macronuclear envelope) through the function of Drp6.I think their findings are highly novel and will provide new insight into a field of cell biology. Especially, their findings will contribute to understanding how specific proteins targeted to the specific intracellular membranes. In addition, their methods (such as floatation assay) for analyzing the interaction between the protein of interest and lipid/liposomes will become an important tool.

We are very happy to note that the reviewer has pointed out the significance of the present study. We fully agree with reviewer and appreciate the thorough analysis and excellent conclusion from the reviewer.

[Editors' note: further revisions were suggested prior to acceptance, as described below.]

Essential revisions:The additional request of one reviewer that should be addressed is to test the involvement of residue 552 and 554 to highlight the significance of isoleucine in position 553 in targeting Drp6 to cardiolipin.

We thank the reviewer for suggesting these excellent sets of experiments. We have now performed the cardiolipin binding and cellular localization of both the mutations at E552 and M554 residues. The results demonstrate that these two neighboring amino acid residues are not essential for providing specificity to bind cardiolipin and for localizing to nuclear envelope. These results clearly suggest that I553 is the key amino acid residue for conferring cardiolipin binding specificity and thereby specificity for nuclear recruitment. We have included these results in Figure 7.